# Deep learning-based segmentation of lithium-ion battery microstructures enhanced by artificially generated electrodes

Simon Müller[1,3], Christina Sauter [1,3], Ramesh Shunmugasundaram[1], Nils Wenzler [1], Vincent De Andrade[2], Francesco De Carlo[2], Ender Konukoglu[1] & Vanessa Wood [1✉]

Accurate 3D representations of lithium-ion battery electrodes, in which the active particles, binder and pore phases are distinguished and labeled, can assist in understanding and ultimately improving battery performance. Here, we demonstrate a methodology for using deep-learning tools to achieve reliable segmentations of volumetric images of electrodes on which standard segmentation approaches fail due to insufficient contrast. We implement the 3D U-Net architecture for segmentation, and, to overcome the limitations of training data obtained experimentally through imaging, we show how synthetic learning data, consisting of realistic artificial electrode structures and their tomographic reconstructions, can be generated and used to enhance network performance. We apply our method to segment x-ray tomographic microscopy images of graphite-silicon composite electrodes and show it is accurate across standard metrics. We then apply it to obtain a statistically meaningful analysis of the microstructural evolution of the carbon-black and binder domain during battery operation.

[1] Department of Information Technology and Electrical Engineering, ETH Zurich, Zurich, Switzerland. [2] Advanced Photon Source, Argonne National Laboratory, Lemont, USA. [3] These authors contributed equally: Simon Müller, Christina Sauter. ✉email: vwood@ethz.ch

The performance of lithium-ion batteries (LIBs) is intimately linked not only to the electrochemical properties of the constituent materials but also to the morphology of these materials[1]. The pore structure of LIB electrodes and separators determines the effective transport coefficient for lithium-ions in the electrolyte[2–4]. Low effective transport increases the ionic resistance, leading to voltage losses (overpotentials), a smaller usable capacity[5], and a reduced rate capability, particularly during fast charge required for automotive applications[6]. In addition, the distribution of the carbon black-binder domain (CBD) around the active particles is crucial to ensure low electronic resistance and mechanical stability throughout battery cycle life[7,8]. The ability to tune structure is of particular importance in emerging applications for LIBs such as electric mobility and grid energy storage that benefit from the uniform operation of hundreds and thousands of cells across their cycle life[9,10].

Accurate 3D representations of the structure in a cell, in which the different material phases are distinguished and labeled (i.e. segmented), aid in the rational selection of materials, manufacturing processes, and operational parameters[11]. While electron- and neutron-based imaging techniques offer specific advantages and can be used together with x-ray-based imaging to provide enhanced and correlated datasets[12–14], x-ray-based tomographic analysis has emerged as a technique of choice, offering a range of resolutions with voxel sizes from millimeters to tens of nanometers and the possibility for non-destructive in situ or operando investigation to monitor the evolution of the internal electrode structure over time[15,16].

However, obtaining 3D reconstructions that can be accurately segmented and quantitatively analyzed is still a challenge, primarily due to (i) the diverging length scales present in LIB electrodes, (ii) the low contrast between key components, and (iii) the low attenuation of carbon-based materials[16]. Active materials such as $Li(Ni,Mn,Co)O_2$ have particle dimensions in the range of $1–10\,\mu m$[17] and contain transition metal elements that provide good contrast during absorption-based imaging and hence a reliable identification of the particles. Graphitic active particles, which make up a large fraction of commercial negative electrodes, do not offer such contrast potentially leading to large errors in segmentation[18]. Furthermore, identification of the polymeric binder domain, which, in case of low-conductivity active particles also contains conductive additives such as nanoscale carbon black, requires not only high contrast (due to the low atomic number of carbon) but also high spatial resolution imaging (due to the nanometer-sized pores and structural features ranging from 5 to $150\,nm$[19–21]). Techniques with high resolution (~10–50 nm voxel edge lengths) have limited fields of view (~$(10–50\,\mu m)^3$) and consequently, only a small sample volume with a limited number of active particles can be imaged quantitatively, which is problematic since analyzing the active material distribution requires a field of view in the range of 5x the largest particle size or ~100 μm to centimeters[5] in order to be representative. Furthermore, high-resolution imaging typically requires long imaging times, often making it prohibitive to obtain statistically relevant data on the electrode scale by imaging many small samples serially[22,23].

In this work, we show how supervised, deep learning can help address the challenges associated with semantic segmentation of high-resolution, volumetric image data of LIB electrodes. While deep learning algorithms have been applied to assess the state of health of a battery[24], to improve the microstructural design[25,26], or to detect defects during manufacturing[27], microstructure segmentation continues to rely heavily on adapted filtering followed by simple thresholding operations[28]. Even though algorithms originating in the field of machine learning are beginning to be used for segmentation[29,30], crack detection[31] and particle detachment[32] to date, deep learning-based approaches are mainly used on high-contrast systems (i.e., cathodes), and algorithms for complete 3D segmentation such as those which are used in medicine to analyze tomographic full-body scans and identify organs[33], have not been applied to semantic segmentation of LIB electrode datasets. Here, we work with the 3D U-Net architecture for semantic segmentation of volumetric image data[34], and show how it can be trained and implemented to segment 3D reconstruction of electrodes into all their different material components (i.e., active particle, binder, pore). For this purpose, we choose volumetric images of graphite-silicon composite negative electrodes obtained with x-ray tomographic microscopy (XTM) (Fig. 1). Even though even higher resolution techniques than XTM exist[35–38], XTM is a popular method to acquire data on LIB components since it simultaneously offers a reasonable field of view, resolution, and collection times. These data encompass many of the key challenges one faces in volumetric image segmentation of batteries, namely multiple heterogeneous material phases with feature sizes ranging from nano- to micrometers and low contrast between different phases, and we show that they cannot be segmented using standard approaches.

A key challenge in employing a convolutional neural network like 3D U-Net for segmentation of volumetric image data is having sufficient and high-quality learning data so that the algorithm can operate fully automated on the dataset of interest. Learning data consist of image pairs: the "input" image and the "output" image, which is the successfully segmented and labeled version of the "input" image. Typically, learning data is real, experimental data that has been segmented and labeled. However, as explained above, no single x-ray imaging technique provides the contrast, resolution, and field-of-view in 3D needed to accurately segment all material phases in an electrode. Multimodal imaging approaches (e.g., combining x-ray absorption contrast tomography with ptychographic x-ray computed tomography[39]) make it possible to achieve segmented and labeled datasets; however, such experiments are difficult and time-consuming. Indeed, we show that the amount of high-quality multimodal image data obtainable during typical experimental beamtimes at a shared synchrotron facility is insufficient for the training of the neural network and leads to only partial segmentation accuracy, as quantified by the Dice score[40].

To overcome this challenge, we propose and demonstrate the benefits of using hybrid learning datasets (Fig. 1), where computationally generated synthetic datasets augment a limited number of real datasets, acquired using multimodal imaging techniques. This approach of enhancing real learning data with computationally generated data has been used for example with face recognition and scene understanding for autonomous driving applications[41,42]. As shown in Fig. 1, we computationally generate a basic electrode structure based on knowledge of the volume percents of different phases and the size and shape distributions of the different particles. We then use some of the high quality, segmented, and labeled data from multimodal imaging as templates for an automatic style transfer algorithm, CycleGAN[43–45], which we use to convert the basic structures into realistically looking microstructures. We then create the corresponding tomographic reconstructions of these artificially generated electrodes, incorporating the effects of the beamline and measurement (e.g., energy, resolution, artefacts, noise). These tomographic simulations and the corresponding synthetic datasets form the input-output data pairs that can be used to train the network together with the real datasets. With this approach of combining real and synthetic training data, we significantly improve the accuracy of the segmentation and labeling of the XTM data into pore space, graphite and silicon particles, and the carbon black-binder domain.

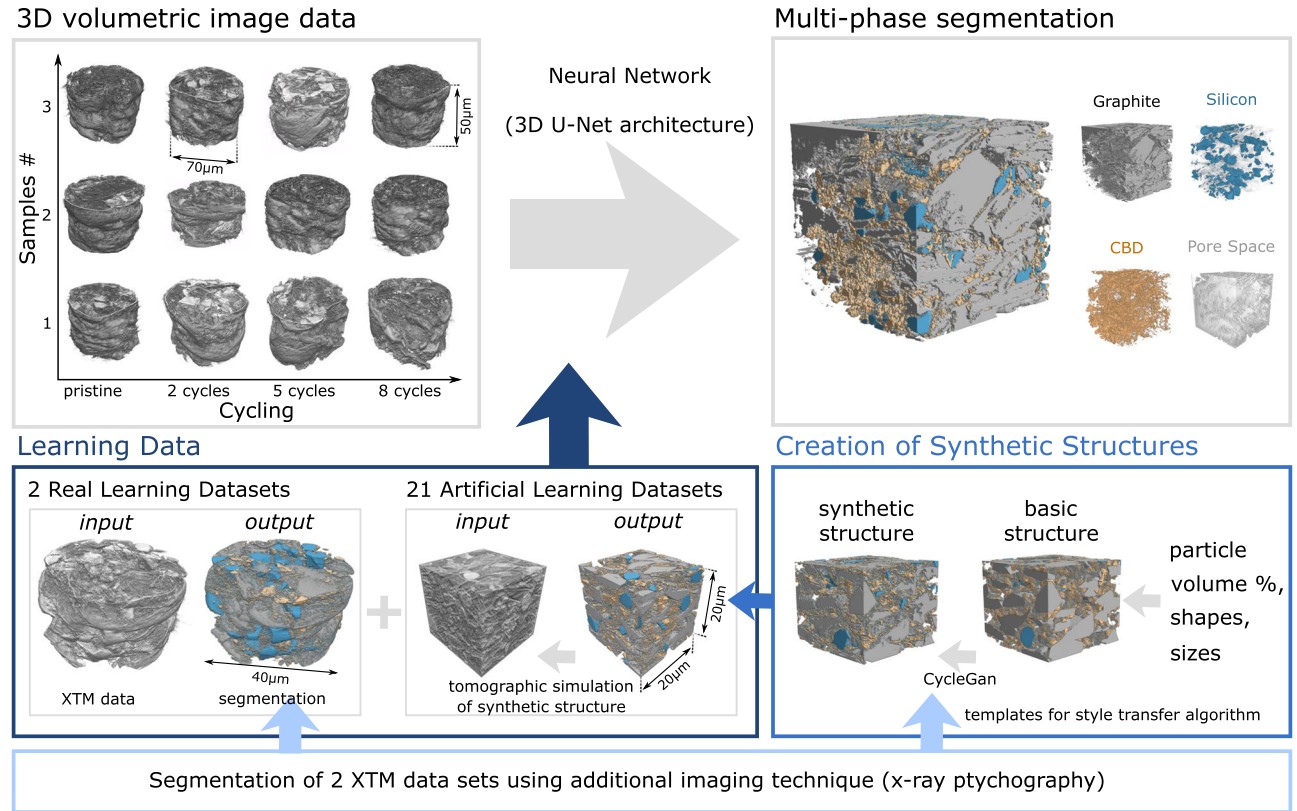

**Fig. 1 Deep learning segmentation of battery electrodes.** The goal of this work is to demonstrate unsupervised, learning-based segmentation of complex volumetric datasets that cannot be easily segmented using standard techniques (e.g., thresholding). We work with pristine and cycled graphite-silicon composite electrodes obtained using x-ray tomographic microscopy (XTM) and aim to segment them into four phases: pore space, graphite particles, silicon particles, and carbon black-binder domain (CBD) for statistical analysis of structural changes as a function of electrochemical cycling. For segmentation, we apply the 3D U-Net neural network architecture, that must be trained using learning data (dark blue box), which consists of volumetric image pairs (i.e., an "input" image like the one to be segmented and a corresponding "output" image, which is a segmented version of the input image). Experimental (i.e., "real") learning data is difficult to obtain and requires multimodal imaging to obtain a segmented output image (see light blue box). Computationally generated (i.e., "artificial") learning data can be added to the real learning data to improve the training of the network and its performance. We generate synthetic structures (blue box) by creating a basic structure based on knowledge of the volume percents and size and shape distributions of the constituent materials and then using real segmented data (light blue box) as templates for an image-to-image translation algorithm (here, CycleGAN) to create a realistic-looking, segmented structure (i.e., "output" image). "Input" images are then generated by simulating how these synthetic structures would look in our XTM measurement based on knowledge of the experimental conditions (energy, resolution, noise).

Having an automated and reliable segmentation tool trained in large part on computer-generated data enables us to address the trade-off between the obtainable resolution and the volume imaged. Specifically, we can work with small sample sizes needed for quantitative imaging at small length scales, but image and segment multiple electrode samples in order to obtain in total a larger volume and therefore statistical insights into the structure of the different material phases at their different length scales. Here, we show that these segmentations can be used to gain insight into the microstructural evolution of these composite anodes as a function of electrochemical cycling. For example, we are able to show that in a graphite-silicon composite electrode most of the morphological changes to the structure of the carbon-black binder domain occur locally around the silicon particles and not in the vicinity of graphite particles.

## Results

**Need for deep-learning based approach.** We image samples of graphite-silicon composite negative electrodes with XTM and aim for an accurate segmentation of the four phases: pore space, graphite, silicon, and carbon black-binder domain.

The samples are taken from a pristine electrode and electrodes cycled twice, five times and eight times. We use ultrashort pulsed laser milling[39] to achieve cylindrical sample diameters below 70 μm. Larger sizes would result in local tomography with a decrease in image quality and an increase in the number and strength of artefacts. We image in total twelve samples, three from each of the cycling conditions, because a sample size of 70 μm is close to the limit of representativeness of electrode structure, with even commercial electrodes exhibiting heterogeneity at this length scale[5]. The projections are reconstructed into image stacks using the ASTRA toolbox[46].

The voxel edge length is 27.5 nm and the spatial resolution is 119 nm according to the Fourier shell correlation (Supplementary Note 2)[47]. Details of the electrode preparation, cycling, sample preparation, imaging, and reconstruction is found in the Methods.

The need for advanced segmentation is clear. A cross-section of an example tomogram is shown in Fig. 2a. Despite the low contrast, leading to an unfavorable gray value histogram (Fig. 2d), manual segmentation is possible (Supplementary Note 5), but is not feasible for rapid analysis of the twelve imaged volumes. As depicted in Fig. 2b, conventional methods like simple threshold-ing (here done with a k-means implementation) do not lead to satisfactory results. This is mainly due to the fact that graphite particles do not exhibit any gray value contrast with respect to the

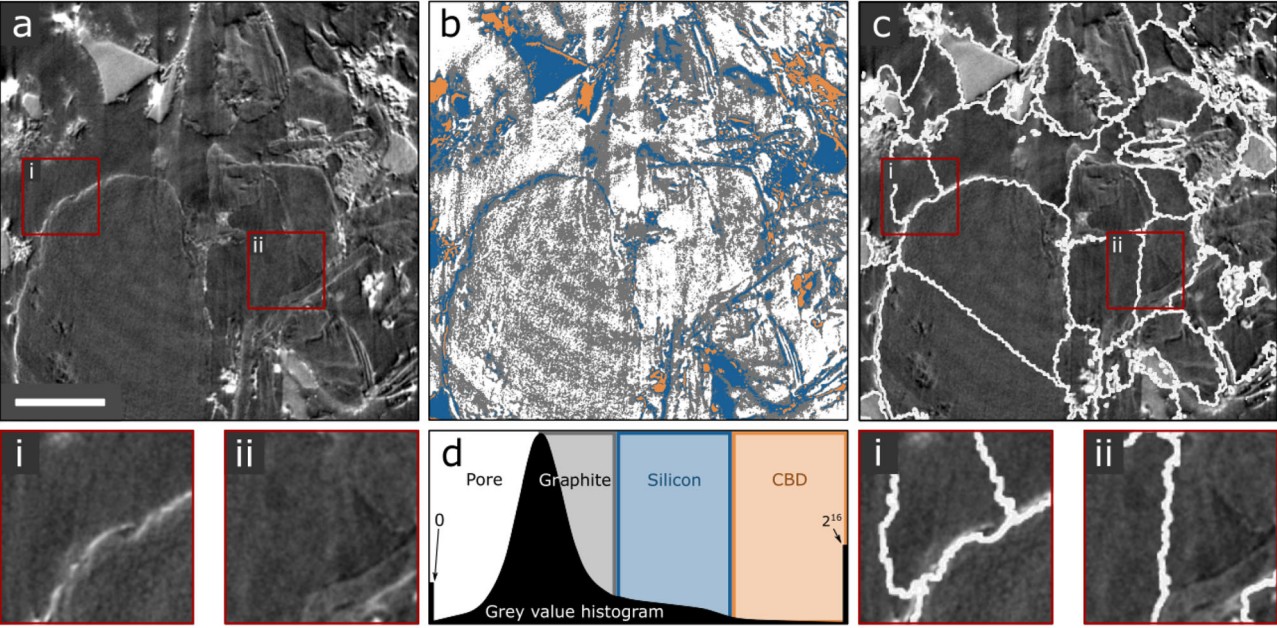

**Fig. 2 Challenging segmentation.** (**a**) A subsection of a tomogram of a graphite-silicon composite electrode imaged with XTM. Scale bar is 5 µm. Neither (**b**) threshold- nor (**c**) random walker-based segmentation yield satisfying results. Low contrast regions are not split up accurately into particle regions (insets i and ii). (**d**) Gray value histogram of the tomogram and the proposed k-means clusters for the pore space (white), graphite particles (gray), silicon particles (blue), and carbon black-binder domain (CBD) (orange).

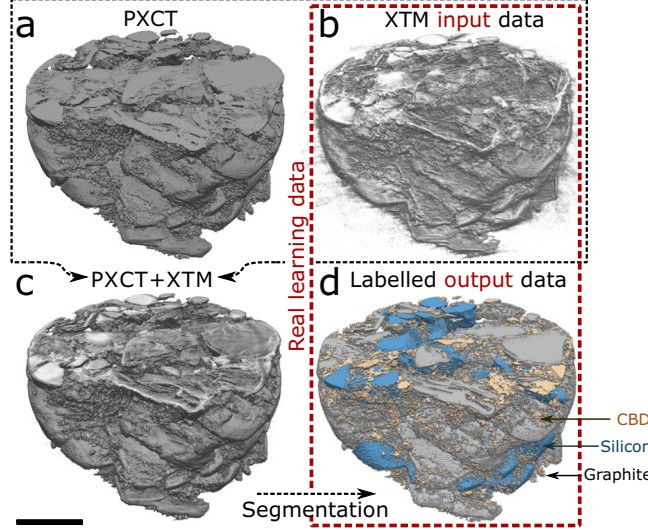

**Fig. 3 Real learning data.** (**a**) Ptychographic x-ray computed tomography (PXCT) and (**b**) x-ray tomographic microscopy (XTM) is performed on the graphite-silicon composite electrodes. Data from these two imaging approaches are (**c**) combined and (**d**) segmented. As indicated by the red box, real training data are made up of input images (the XTM images in (**b**)) and output images (the segmented, multimodal images in (**d**)). Scale bar is 10 µm.

pore space and their contour can only be guessed based on the bright edge-enhancement effect. Graph-based processing (using a 3D implementation of the random walker algorithm[48]) (Fig. 2c) is better able to identify particle boundaries but fails in low contrast regions, such as inside graphite particles where particle boundaries are erroneously found (see Fig. 2, insets i and ii). In addition, such a random walker algorithm requires the input of seed points (i.e., the particle center locations), which is a challenge and often requires human verification. While classical segmentation procedures are not sufficient for an accurate

segmentation of the four phases, the fact that the human eye can pick out different phases suggests that a learning-based approach to semantic segmentation would be applicable to the datasets.

**Deep learning-based segmentation.** We, therefore, turn to deep learning algorithms, which have entered the field of volumetric image segmentation through the implementation of 3D convolutional networks[49]. Prominent architectures that apply these principles for volumetric segmentation are the V-Net[50] and, more widely used, the 3D U-Net[34]. We borrow a specific implementation of the 3D U-Net architecture from Baumgartner et al. that was demonstrated to work well on volumetric medical images captured with magnetic resonance imaging (MRI) [51]. Like these images, our electrode image datasets are three-dimensional and comparable regarding gray value contrast and the number of distinguishable material phases.

This network must be trained. The approach to obtain real learning data is shown in Fig. 3. In a previous study[39], we showed how the same electrode sample can be imaged with ptychographic x-ray computed tomography (PXCT) (Fig. 3a) and XTM (Fig. 3b), and how this data can be combined (Fig. 3c) and used to achieve precise segmentation based on gray value thresholding (Fig. 3d). We have two cylindrical volumes, each 40 µm in diameter and 20 µm in height, one from a pristine and one from a cycled graphite-silicon composite electrode, that have undergone this multimodal imaging and segmentation procedure. The XTM reconstructions are the "input images" and their segmentations are the "output image" label maps that make up the learning data.

Due to memory limitations, 1000 subvolume pairs (the 3D image data and label map) of dimensions $15.6 \times 15.6 \times 3$ µm ($256 \times 256 \times 48$ voxel) are randomly sampled (and thus possibly overlapping) from the learning data such that the smallest dimension is equally likely to be along the $x$, $y$, or $z$ axis. 50% of the training data is subjected to data augmentation based on basic image manipulations in order to render the neural network more robust and increase its ability to generalize while decreasing the

Segmentations achieved using

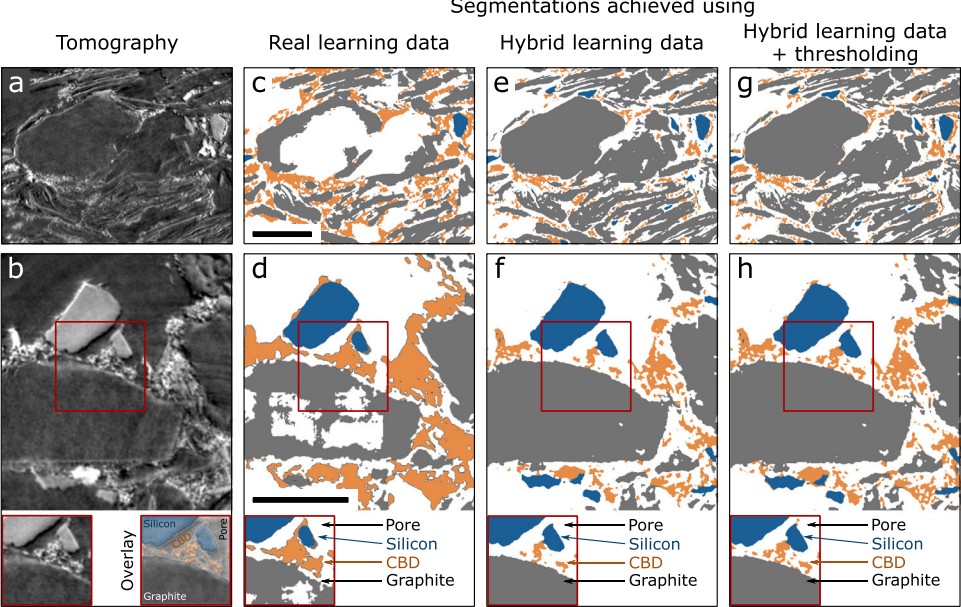

**Fig. 4 Benefits of artificial learning data for segmentation.** A through-plane cross-section (**a**) and in-plane cross-section (**b**) of the raw data are shown in order to demonstrate the segmentation achieved using different training datasets. When trained only on the limited real learning data (**c**, **d**), the neural network correctly identifies the silicon particles but fails to reliably distinguish pore space from graphite particles. When trained with the hybrid learning data (**e**, **f**), the neural network is better at identifying graphite from pore space. Taking the graphite and silicon phase predicted by the neural network and adding the carbon black-binder domain (CBD) resulting from thresholding (**g**, **h**) improves the fine details of the CBD. The insets in (**b**, **d**, **f**, **h**) highlight the improvement of the segmentation, especially for the graphite and CBD domain. The overlay in (**b**) shows the final segmentation result from (**h**) over the original tomographic image. Scale bars are 10 μm.

potential danger of overfitting[51]. Specifically, we apply Gaussian noise and blurring, vary contrast and brightness, and use affine transformations. The model is trained for 100 epochs, which amounts to a total training time of 72 h on a Nvidia TITAN Xp GPU.

The results of the segmentation of XTM images are shown in Fig. 4. Comparing cross-sections (Fig. 4a, b) of the raw image data and the segmentation achieved for these cuts (Fig. 4c, d), we see that the neural network trained on real learning data fails to reliably distinguish pore space from graphite particles, fails to properly segment the fine features of the carbon black-binder domain, and confuses clusters of the carbon black-binder domain for silicon, particularly at particle boundaries.

To quantify the segmentation quality, 400 sequential slices of the XTM dataset of one of the three pristine samples are manually segmented with the help of the Dragonfly software. This manually segmented volume serves as the "ground truth" and is otherwise not used for training. We evaluate segmentation quality according to nineteen standard metrics based on similarity and distance criteria[40], where the learning-based segmentation of one pristine sample are compared to the corresponding "ground truth" (Table S2).

We focus on the Dice coefficient[40] (Table 1), which is the normalized volumetric overlap of voxels in the segmentation and the "ground truth" of a given phase. A Dice score of zero means no overlap between segmentation and the "ground truth"; a Dice value of one corresponds to complete overlap between segmentation and the "ground truth" segmentation. For the network trained on the real learning data, Dice coefficients between 0.6 and 0.7 are found for the active particles and pore space, while the Dice coefficient for the carbon-black binder domain is only 0.38. This is consistent with our findings from visual inspection (Fig. 4).

While the learning-based segmentation is a clear improvement over simple thresholding or random-walk-based segmentation approaches (Fig. 2), we attribute the less than satisfactory segmentation to the limitations of the real learning datasets.

**Table 1 Segmentation quality.**

| Dice values | Pore space | Graphite | Silicon | CBD |
|---|---|---|---|---|
| Real learning data | 0.63 | 0.65 | 0.69 | 0.38 |
| Hybrid learning data | 0.69 | 0.77 | 0.82 | 0.58 |
| Hybrid + thresholding | 0.72 | 0.77 | 0.82 | 0.72 |

Dice values quantifying the quality of the segmentation for the pore space, the graphite particles, the silicon particles, and the carbon black-binder domain (CBD). The Dice values compare the neural network segmentation (trained on real learning data, hybrid learning data, or hybrid learning data together with thresholding) with a manual segmentation of one of the pristine electrode samples.

For example, in the real dataset, large graphite particles may be cut in the sample preparation prior to the XTM and PXCT imaging, which means that the neural network will tend to underestimate particle sizes and thereby misattribute the portions of graphite particles to pore space.

Simply adding to real learning data by taking more high-resolution images is impractical from a time and resource perspective. Each high-resolution PXCT scan takes 50 h. Furthermore, we have recognized that the limited sample volume leads to challenges in terms of the representative nature of the real learning dataset.

For this reason, we turn to computationally generated (i.e., "synthetic" or "artificial") learning data, which offers control over the data content (i.e., particle sizes and volume fractions) such that the most common segmentation failures (e.g., misinterpreting pore space for graphite) are extensively trained and the neural network is more robust.

**Generation of synthetic learning data.** To create realistic artificial data that have the characteristics of the experimental image

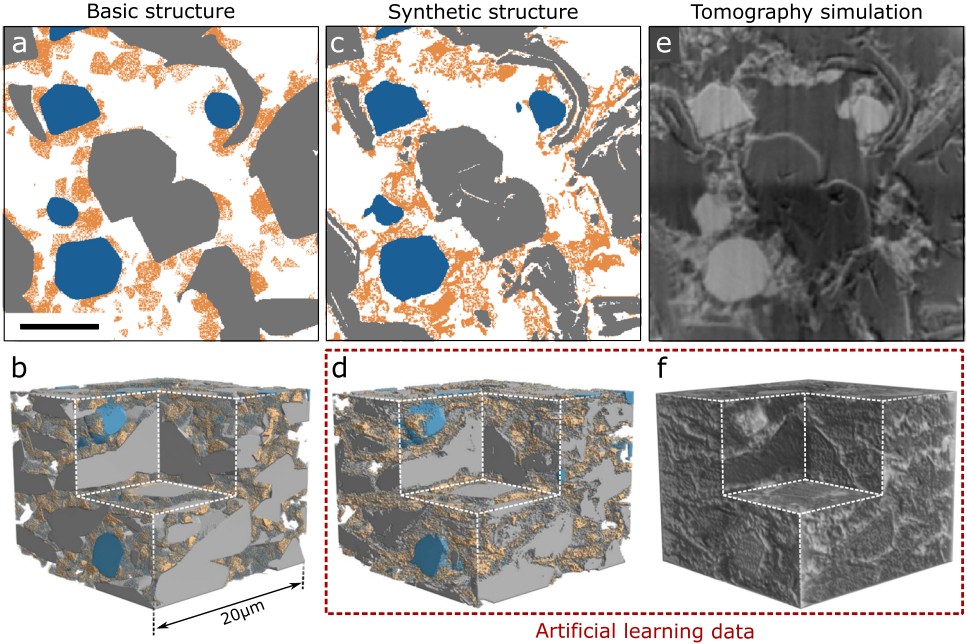

**Fig. 5 Generation of artificial learning data.** An initial basic structure is generated based on polygon shapes as shown in 2D (**a**) and 3D (**b**). Using a style transfer algorithm, a synthetic structure is achieved (**c, d**). Tomography simulations of the synthetic structure (**f**), the cross-section shown in (**e**), complete the generation of artificial learning data. The scale bar for (**a, c**), and (**e**) is 5 μm.

data which are labeled, we begin by generating a basic electrode structure (Fig. 5a, b). This can be done with dedicated tools (e.g., Math2Market) or with MATLAB as in the present case with details found in Supplementary Note 3. Particle volume fractions, particle sizes, shapes, and orientations are defined for the graphite, silicon, and the carbon black-binder domain phases. One could additionally implement physics-based approaches[52,53]. In order to serve as training data, these basic electrode structures have to be further refined to include the characteristic shapes and typical small-scale heterogeneity of the carbon black-binder domain. This is achieved by applying a style transfer computed by the CycleGAN algorithm[43–45] with the real segmented structures serving as templates (Supplementary Note 3). The resulting computationally generated electrode structures are referred to as synthetic structures (Fig. 5c, d).

To form a complete training dataset, the labeled synthetic structures need corresponding volumetric images that must resemble the collected tomographic projections (i.e., intensity images). This can be done with the ASTRA toolbox in MATLAB[46], which allows 2D projections to be calculated from arbitrary structures. Literature values of the refractive indexes are assigned to the respective material phases of the synthetic structures and projections are calculated (see Supplementary Note 4) and afterwards reconstructed the same way that the real tomography data are reconstructed. While simulating the x-ray tomography, special care is taken to include a locally varying background illumination and a spatially changing amount of edge-enhancement, as can be seen in Fig. 5e. The resulting simulated tomography images (Fig. 5f) together with the labeled synthetic structures form the artificial learning datasets.

**Deep learning segmentation with synthetic and real learning data.** Using 21 artificial learning datasets (20 μm × 20 μm × 20 μm in size) in addition to the two real datasets (cylinders of 40 μm diameter and 20 μm height), the neural network models are again trained. The procedure for training is kept the same as before, except that, this time, 500 volume pairs are sampled from

real learning datasets and 500 volume pairs from artificial learning datasets. We refer to this as hybrid learning data.

The benefits of adding artificial datasets to the learning data are evident in the quality of the segmentation shown visually in Fig. 4 as well as the quantification (Table 1 and Supplementary Table 2). All metrics show the same trend, highlighting how the segmentation quality improves if the neural network is trained on a mix of hybrid datasets instead of a limited amount of real datasets.

For segmentation of pore space, training with hybrid learning data instead of real data leads to improvement of the Dice coefficients from 0.63 to 0.69. For segmentation of graphite, the improvement is from 0.69 and 0.77. These improvements are due to the fact that, by training with the hybrid datasets, the network no longer mistakes the inside of a graphite particle for pore space (compare Fig. 4d and f). We believe that this occurs because artificial datasets are designed to be representative of the volume percents of each phase and the particle size distributions and contain more large graphite particles than the real datasets, which are often randomly cut in the sample preparation prior to imaging. Nonetheless, segmentation of the graphite and pore phases remains a challenge because of the low contrast between the graphite and pore space, the lack of distinguishing features within bulk graphite, and imaging artefacts around the edges of small graphite particles. Furthermore, the size of the receptive field[54] (i.e., the subset of the input image that influences the prediction for a certain voxel), is at times smaller than a graphite particle.

Training with hybrid learning data also allows the carbon black-binder domain clusters at the edges of silicon particles to be more accurately distinguished (compare Fig. 4d and f), with the Dice coefficient for the carbon black-binder domain improving from 0.38 (training of the model with real data only) to 0.58 (training with hybrid data) and the Dice score for silicon improving from 0.69 (only real training data) to 0.82 (hybrid learning data). This improvement can be attributed to the inclusion of different illuminations and edge enhancements in the artificial datasets that are reflective of the variations present at the

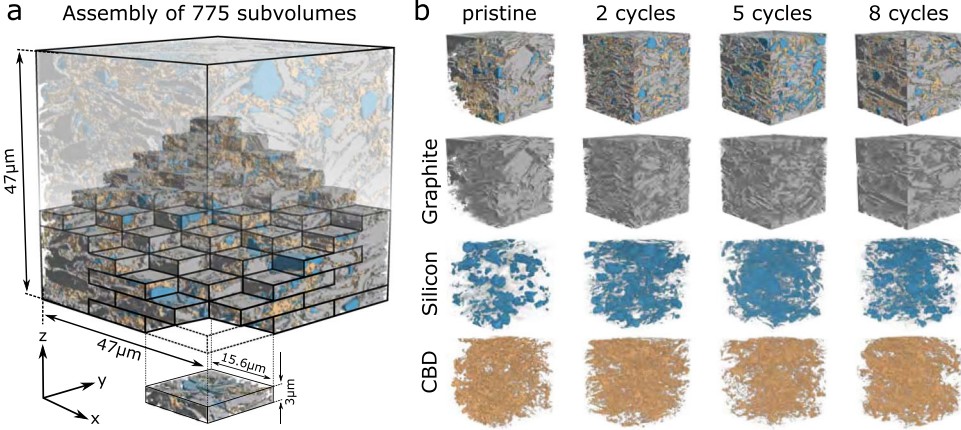

**Fig. 6 Deep Learning Segmentation.** Illustration highlighting that for each sample, the deep learning approach is applied to 775 subvolumes (**a**), which, once segmented, are reassembled. The multiphase segmentation (**b**) for pristine samples and samples cycled twice, five times, and eight times enable a number of different microstructural analysis.

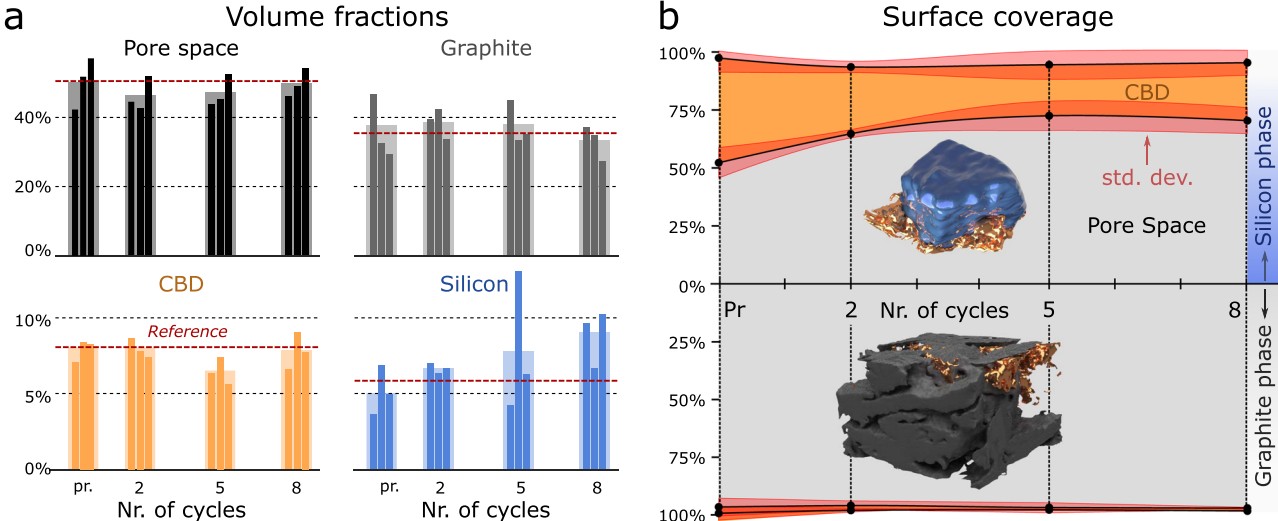

**Fig. 7 Analysis of the segmented electrodes.** Volume fractions (**a**) occupied by pore space (black), graphite particles (gray), carbon black-binder domain (CBD) (yellow), and silicon particles (blue) for the three pristine (pr.) and the three samples cycled two, five, or eight times. The dashed red line indicates the average volume fraction expected for each phase based on the electrode manufacturing. Surface coverage (**b**) of the graphite particles (lower) and silicon particles (upper) with pore space (gray) and CBD (orange) over the different cycle states. The shaded regions indicate the standard deviations.

beamline but that are not fully captured in the limited amount of real training data.

To further improve the accuracy of the carbon black-binder domain and pore space segmentation, we can add basic thresholding on top of the neural network prediction. For each of the twelve tomograms, we determine an intensity threshold from the gray value histogram corresponding to the boundary between what is identified as graphite and silicon particles, as determined by a k-means approach requiring no manual interaction (see for example Fig. 2d). We then apply this threshold to all voxels identified as carbon black-binder domain and pore space by the neural network. Since the carbon black-binder domain contains copper nanoparticles as a staining element that has higher x-ray absorption than silicon, we assign voxels with a gray value above the threshold to the carbon black-binder domain and voxels below the threshold to the pore space. This procedure improves the segmentation of the smaller structural features of the carbon black-binder domain (Fig. 4h) and leads to a Dice coefficient of 0.72. While the shape of carbon black-binder domain clusters is now segmented, FIB-SEM imaging indicates that the carbon black-binder domain

has an internal porosity of $27 \pm 3\%$[55]. However, this nanoscale porosity internal to the carbon black-binder domain is below the image resolution (119 nm) and therefore is not visible or segmentable.

There are no previous Dice scores reported for the segmentation of battery electrodes so the values achieved here cannot be compared to other segmentation approaches. Although a detailed comparison is difficult as four-phase segmentations on this scale with similar complexity and image quality are hard to find, the achieved Dice scores lie in the range of comparable medical studies[56].

**Segmentation of electrode microstructures**. We thus apply a neural network model trained on the hybrid learning data with subsequent thresholding to segment all twelve XTM datasets (see Fig. 1). The tomographic reconstructions are binned by a factor of 2.22, resulting in a voxel size of 61 nm. Cubes of 47 μm edge length (768 voxels) are cut from the cylindrical volumes for segmentation and analysis. Because of memory limitations, the volumes cannot be segmented at once but have to be divided into

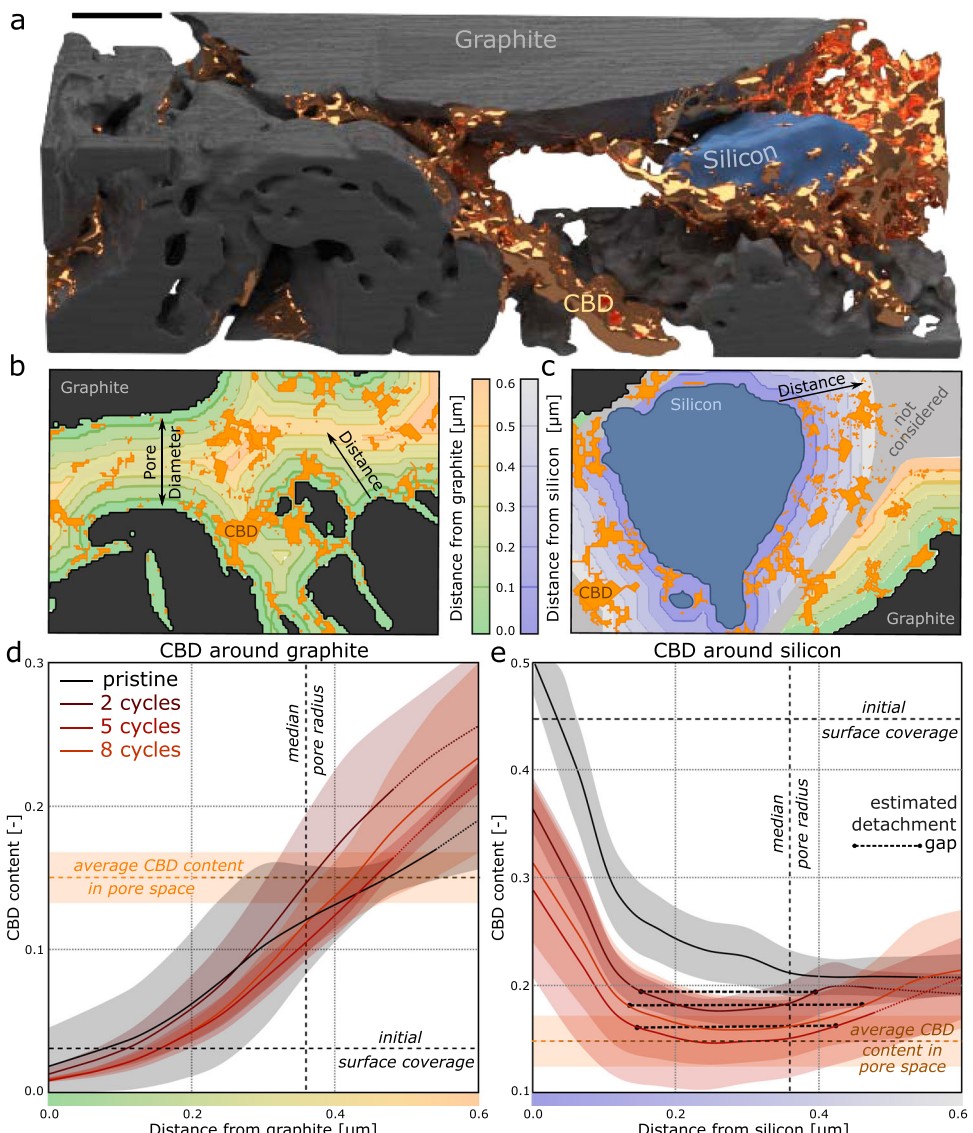

**Fig. 8 Evolution of microstructure with cycling.** Reconstruction of a subvolume (**a**) of a pristine electrode showing how the carbon black-binder domain (CBD) (gold) clusters in pores, particularly around silicon particles (blue). Illustration showing how concentric shells around the graphite (**b**) and silicon phases (**c**) are defined. The CBD content in shells emanating around graphite (**d**) and silicon (**e**) particles for the pristine sample (black) and for samples cycled two, five and eight times. The shading represents the standard deviation of the three samples. Near silicon particles, the positioning of CBD changes indicating detachment and gap formation (dashed lines) upon electrode cycling. Scale bar is 2 µm.

subvolumes of 48 × 256 × 256 voxels that are then processed by the neural network. Due to the anisotropy in processing volume dimensions, each tomographic dataset is processed independently in x (subsampling with 48 × 256 × 256 voxel volumes), y (subsampling with 256 × 48 × 256 voxel volumes), and z (subsampling with 256 × 256 × 48 voxel volumes) direction. Figure 6a schematically illustrates the process for reassembling the entire z direction. The subvolumes are sampled to overlap by half their edge length, which results in 775 patches per sampling direction to be processed. For each subvolume, a softmax representation is created, which corresponds to a map, where each voxel contains the probability of it belonging to each material phase (pore space, graphite, silicon, or carbon black-binder domain). As described in Supplementary Note 6, the softmax representations from each sampling are then combined to obtain the final segmentation (Fig. 6b).

Based on the electrode fabrication protocol, we expect electrodes to consist on average of 51% pore space, 35% graphite,

6% silicon, and 8% carbon black-binder domain (dashed red lines in Fig. 7a). Considering the limited sample size used for imaging, which leads to variations in the phase compositions over different samples, the volume fractions determined from the segmentation (49% pore space, 36% graphite, 7% silicon, and 8% carbon black-binder domain) match well with respect to the reference values.

**Microstructural analysis and changes with cycling.** The segmentation achieved here provides new opportunities for analyzing the microstructure of the graphite-silicon composite electrodes and its morphological evolution during electrochemical cycling. We note that because this is an ex-situ study, we are not tracking the microstructural changes of the same sub-section of electrode with cycling. Furthermore, given the small sample size and the resulting compositional variations across the three samples (Fig. 7a), we, therefore, focus on trends that hold for all three samples and detailed analysis of changes around phases (e.g., the

**Table 2 Parameters pertaining to lithium transport in pore space.**

| | | Number of cycles | | | |
|---|---|---|---|---|---|
| | | Pristine | 2 | 5 | 8 |
| Effective transport coefficient [-] | Including CBD | 0.1492 ± 0.0369 | 0.1277 ± 0.0287 | 0.1351 ± 0.0292 | 0.1435 ± 0.0224 |
| | Neglecting CBD | 0.2098 ± 0.0445 | 0.1853 ± 0.0316 | 0.1810 ± 0.0308 | 0.2004 ± 0.0307 |
| Porosity [%] | Including CBD | 50.52 ± 7.57 | 46.58 ± 4.86 | 47.52 ± 4.67 | 50.04 ± 4.09 |
| | Neglecting CBD | 58.47 ± 8.25 | 54.58 ± 4.46 | 54.00 ± 4.05 | 57.88 ± 4.62 |
| Tortuosity in through plane direction [-] | Including CBD | 3.3465 ± 0.3656 | 3.6076 ± 0.5970 | 3.4711 ± 0.4273 | 3.4092 ± 0.2334 |
| | Neglecting CBD | 2.7651 ± 0.2209 | 2.9287 ± 0.4523 | 2.9587 ± 0.3465 | 2.8558 ± 0.2261 |

Mean effective transport coefficients, porosity, and tortuosity values for the three different samples per cycling state under consideration and negligence of the carbon black-binder domain (CBD) in the pore space.

changes in the carbon black-binder domain around graphite or silicon active particles).

An interface between an active particle and the carbon black-binder domain facilitates electronic connectivity to the particle, while an interface to pore space (which is infilled with electrolyte) will see the largest mass (i.e., lithium-ion) transport. The location and extent of the different interfaces play an important role in the lithiation behavior, particularly at fast C-rates[57]. The evolution of the surface coverage of the graphite and silicon particles with carbon black-binder domain over repeated cycling is shown in Fig. 7b. The lower panel of Fig. 7b shows that 97% of the surface of graphite particles is in contact with the pore phase, while only 3% is in contact with the carbon black-binder domain. This value may be an underestimate since the identification of graphite particles during segmentation is based on edge-enhancement of the tomography data (Fig. 2a), making it difficult to distinguish bright graphite edges and the bright carbon-black and binder domain signal. At the same time, even in regions where graphite surface is in contact with the carbon black-binder domain, there is internal porosity to the carbon black-binder domain that is not resolved with this imaging. Importantly, however, these values do not change after cycling, highlighting that cycling does not induce a permanent change in the morphology of the carbon black-binder domain in the vicinity of a graphite particle. In contrast, the surface of pristine silicon particles is 53% in contact with pore space and 46% is covered by the carbon black-binder domain (Fig. 7b, top panel). Even within the first two cycles at the slow rate of C/20, the silicon surface in contact with pore space increases to 65%, suggesting that the carbon black-binder domain detaches from the silicon particles, which is consistent with the previous work[22]. During the first cycle (formation cycle), a solid electrolyte interface (SEI) forms on the surface of the silicon particles[58]. Then, due to the large volumetric changes of silicon upon lithiation and delithiation, the carbon black-binder domain connected to the silicon experiences stress and detaches at those locations where adhesive forces are insufficient.

Another important parameter linking electrode microstructure to battery performance is the effective transport coefficient, $\delta$, which is defined as the ratio of the diffusivity in the electrolyte-filled electrode, $D_{eff}$, to the diffusivity in the bulk electrolyte, $D$ ($\delta = \frac{D_{eff}}{D}$). The effective transport coefficient can also be linked to the tortuosity $\tau$ and the porosity $\varepsilon$ of the electrode ($\delta = \frac{\varepsilon}{\tau}^2$). We perform numerical diffusion simulations on the electrode structures and find that the effective transport coefficient is 0.14. However, given the relatively small field of view, this value should be considered with caution, as uncertainty due to lack of representativity is expected to be high.

The small electrode subvolume in Fig. 8a visually highlights the importance of the carbon black-binder domain morphology on the lithium transport in the pore space, and we can use the

multiphase segmentation to determine to what extent the carbon black-binder domain contributes to the low effective transport. The effective transport coefficient neglecting the carbon black-binder domain (i.e. considering the carbon black-binder domain as part of the pore phase), is 0.19. This means that the carbon black-binder domain decreases the effective transport by an additional 27% (i.e., the diffusivity of lithium in the pore is decreased from 19% to 14%). As listed in Table 2, this decrease in effective transport is caused by both a decrease in porosity and an increase in tortuosity. This is in agreement with previous studies, which found up to 1.5 times higher effective transport values measured via electrochemical impedance spectroscopy (EIS) compared to values calculated numerically on electrode structures acquired with x-ray tomography that only resolved the active particle phase[59]. While the impact of the carbon black-binder domain on transport is clear, when using segmented 3D images to compute the effective transport coefficient, it is important to keep in mind discrepancies between numerical diffusion simulations and EIS measurements[60] as well as incomplete pore infilling[61]. Nonetheless, quantification of the morphology of the carbon black-binder domain is one step towards understanding ionic and electronic performance of the electrode and it can be used to form the basis of computer-generated carbon black-binder domains[22,52,55,57,62,63]; however, the specific transport properties of ionic and electronic transport of the carbon black-binder domain must also be better understood.

Interestingly, the calculated effective transport coefficient stays constant as a function of cycling (Table 2), suggesting that the morphology of the carbon black-binder domain within the pore space does not change significantly despite the fact that we know there are dynamic changes happening to the pore space of the electrode during cycling[16] and our segmented images show that the carbon black-binder domain detaches from the silicon particles. To confirm this surprising lack of change in the effective transport coefficient, we consider other morphological descriptors (e.g., distribution of pore radius size, distribution of the size of the carbon black-binder domain features) as described in Supplementary Note 7. To eliminate the possibility that the different volume fractions of the phases in the different samples make sample-to-sample comparison difficult, we look for trends with cycling in sub-volumes of the electrodes with similar volume fraction of the different phases. Again, trends in the morphology of the carbon black-binder domain are not observed at the electrode level (Supplementary Fig. 8b). However, the volume fraction of silicon in the electrode is small (6%), meaning that the detachment of the carbon black-binder domain from the silicon particle, which likely leads to the greatest change in carbon black-binder domain morphology, is found only in small regions of every sample. Trends induced by the cycling behavior of the silicon can easily be hidden within the statistical variation found

sample to sample, and would require a more and/or larger volumes to be analyzed in order to account for the variability.

We, therefore, use our segmented reconstructions to perform location-specific structural analysis. We define volume shells (each 61 nm wide) radially around the graphite particle phase and the silicon particle phase (shaded regions in Fig. 8b, c). Volumes formed by overlapping shells around graphite and shells around silicon are not considered (gray shaded area in Fig. 8c) since effects in such a region cannot be robustly assigned to either graphite or silicon. Finally, regions that are further away than 600 nm since effects occurring in such regions again are difficult to assign due to the influence of graphite or silicon.

We analyze the volume fraction of carbon black-binder domain within each shell around graphite (Fig. 8d) and around silicon (Fig. 8e) for pristine (black) and cycled samples (dark red to orange with increasing cycling). The $x$-axis indicates the distance from the graphite or silicon surface, where the median pore radius in the electrode (360 nm) is indicated. As expected, in the volume shell at the graphite surface, the carbon black-binder domain content matches the surface coverage of graphite of 3% (Fig. 7b). The increase in the amount of carbon black-binder domain in subsequent volume shells (Fig. 8d) can be explained by the clusters of carbon black-binder in the pore space that bridge between active particles, which also becomes apparent during visual inspection. Electrochemical cycling of the samples does not influence the distribution near the surface as seen by the overlap of the lines within the error (shaded regions), but does tend to compress the carbon black-binder domain closer to the middle of the pore space.

Analysis of the carbon black-binder domain around silicon (Fig. 8e) shows that the silicon surface is initially up to 50% covered by the carbon black-binder domain and this amount decreases to 40% upon cycling, consistent with the analysis in Fig. 7b. For the pristine sample, the carbon black-binder domain content decreases with increasing distance to the silicon phase until it converges to around 20% (the average percentage of carbon black-binder domain in the pore space is 17%). In the cycled samples, the carbon black-binder domain content reaches a minimum at a distance of roughly 300 nm from the particles before increasing again. This is indicative of regions of the carbon black-binder domain that have detached from the silicon particle having moved into the middle of the pore, and is consistent with previous reports finding a gap between the silicon and carbon black of ~250 nm[22].

The statistical analysis of structure enabled by our segmented reconstructions leads to several findings. First, the distribution of the carbon black-binder around graphite particles and around silicon particles differs, perhaps due to differences in the adhesion of the carbon black-binder to the two-particle types or the local differences in the slurry mixing (e.g., shear forces). Such different distributions certainly impact the mechanical and electrochemical properties of the electrode, and this serves as an important reminder when computationally generating carbon black-binder structures for simulation purposes that, in composite electrodes, the carbon black-binder domain may look structurally different in sub-regions. Second, for our electrode, the structural changes in the carbon black-binder during electrochemical cycling do not lead to a resolvable change in the pore size distribution or tortuosity on the characteristic length scale of the electrode (~μm range); however, we do find local changes, particularly around the silicon active particles. These changes do not involve a break or large distortion of the carbon black-binder network (which would be visible as a change in pore size and a change in tortuosity), but rather lead to partial detachment of the carbon-black binder domain from active particles and repositioning of the carbon black-binder within electrode pores. Such detachment and repositioning may cause changes to the pore structure (e.g.,

shape and tortuosity) internal to the carbon black-binder domain but such changes are not visible with the resolution of 119 nm of this experiment. Finally, we should recall that these findings on the structure and structural evolution are unique to each electrode composition and its cycling protocol. More dramatic morphological changes might be observed in other types of electrodes or at other cycling rates. Also, since our analysis looks at the electrodes between cycling steps, our conclusions are about permanent deformation and not temporary fluctuations to the microstructure that occur during cycling. With the help of more accurate representations of the microstructure, the dynamical changes due to mass transfer can now hopefully be better structured.

## Discussion

Even state-of-the-art volumetric imaging of lithium-ion battery electrodes only provides suboptimal image quality due to technical limitations or practical constraints (e.g., time or system availability). Accurate segmentation of volumetric datasets is difficult, and while deep learning-based segmentation can help, its effectiveness is dependent on the availability of high-quality learning data that is in many cases not readily available.

Our investigation of the carbon black-binder domain in a graphite-silicon composite anode after cycling shows the feasibility and value of synthetic data. Indeed, our work highlights how lithium-ion battery electrodes present an interesting use case for deep learning-based segmentation on synthetic generated datasets. In contrast to medical imaging[64], for example, where significant unknowns exist in the type of features that may be found in volumetric image data, a LIB cell manufacturer knows the volume fractions of the different material phases, the particle size, and shape distributions. Post-mortem analysis[65] can be carried out on cells to provide information on the type of features to be expected following cycling. Together with understanding of the imaging process, this ground truth information can be used to create synthetic structures and simulate the image data. By populating databases with real and synthetic training data, the community will be able to improve the applicability of deep learning-based algorithms to semantic segment battery electrodes.

Furthermore, in addition to semantic segmentation, tools to make higher-level assignments without segmentation such as directly extracting certain material phases (e.g., carbon black-binder domain network identification), finding regions with defective structure properties (e.g., fractured particles or areas with unusually high or low porosity), or assessing the state of health based on the microstructural morphology should also be pursued.

This work highlights that in addition to the potential usage of machine learning in batteries for materials discovery[66] and prediction of failure[67], there is significant potential to capitalize on algorithms developed in the computer vision space to analyze the chemistry and structure within batteries and its time-dependent evolution during cycling. The ability to gather and quantify large datasets with a well-defined error in their analysis are key to moving past trial-and-error based materials and cell design and assessment, and to enabling higher-level battery model and simulation verification.

## Methods

**Electrode preparation**. The electrode slurry is prepared by mixing 75 wt.% graphite (SLP 30 Timcal) with 10 wt.% silicon (BASF, SiO$x$, $x \approx 1$), 10 wt.% poly-vinyldiene difluoride (PVDF) binder (Kynar Flex® HSV900), and 5 wt.% carbon black (Timcal Super C60, Imerys). To improve the imaging contrast, 50 vol.% of the carbon black in the carbon black-binder domain is replaced by copper nano-particles (US Research Nanomaterials, Inc.)[8,22]. The slurry is coated on copper foil with a doctor blade (150 μm blade gap), and dried overnight at 80 °C with nitrogen flow. 13-mm disk-shaped electrodes are punched out and compressed with 1 t. Further processing takes place in an Argon-filled glovebox.

**Electrochemical cycling**. A half-cell containing electrode, a 250 μm thick glass fiber separator (Whatman® glass microfiber filter), 500 μL standard LP50 electrolyte (BASF, 1 M LiPF6 in ethylene carbonate: ethyl methyl carbonate = 1:1 by weight) and lithium metal as a counter electrode (Alfa Aesar, lithium foil, 99.9%) is assembled in an Argon-filled glovebox. The cells are cycled galvanostatically in a potential range of 10 mV–1.5 V at a rate of C/20 for 2, 5, and 8 cycles with VMP3 battery cycling system (Biologic). Each protocol ends with a 20 h period of constant voltage (3 V).

**Sample preparation for imaging**. Samples with a diameter of 1 mm are punched out of the electrode and glued to a custom-made invar sample holder. Laser milling reduces the sample diameter to below 70 μm.

**XTM measurements**. All XTM measurements are recorded at the 32-ID-C beamline of the Advanced Photon Source at the Argonne National Laboratory at a beam energy of 8.4 keV. 1210 projections per tomographic scan are acquired with an exposure time of 2 s resulting in a measurement time of ~40 min. The 2448 × 2448 pixel detector leads to a voxel size of 27.5 nm.

**Reconstruction**. The ASTRA toolbox[46] in MATLAB is used to filter and reconstruct the acquired projections with the Filter Back Projection (FBP) algorithm. Paganin phase retrieval[68] was performed with a ratio of the refractive index parameters $\delta/\beta = 0.028$ and $\delta/\beta = 0.28$.

**Manual segmentation**. Manual Segmentation is performed using the Dragonfly software. More details can be found in Supplementary Note 5.

**Generation of artificial learning data**. Artificial learning data are generated using MATLAB, the style transfer algorithm CycleGAN and the ASTRA toolbox to simulate the tomography. Details are described in Supplementary Note 3 and Note 4.

**Renderings**. Images and renderings are produced with Arivis, ImageJ, and Inkscape.

**Structural analysis**. The analysis of the 12 segmented datasets is performed in MATLAB using code from Legland et al.[69]. The diffusivity in the through-plane direction was calculated on volumes scaled by a factor of 0.5 (384 × 384 × 384 voxel) using the DiffuDict toolbox of the GeoDict2020 Software (Math2Market GmbH, Kaiserslauten, Germany) applying symmetric boundary conditions (Dirichlet). Both border planes (along the through-plane direction) are set to have a constant concentration which decreases edge effects. Symmetric boundary conditions are also applied in all tangential directions and the Neumann (zero flux) condition is used at the domain boundaries. The concentration drop across the through-plane direction was set to 1 and the (dimensionless) Laplace equation is solved in an iterative process by finite volume-based solver (EJ) and stops if the process becomes stationary (tolerance of 0.001).

## Data availability

The acquired x-ray tomography (XTM) data along with their respective labeled data have been deposited in the ETH Zürich Research Collection and are available under https://doi.org/10.3929/ethz-b-000505935. The ptychographic x-ray computed tomography (PXCT) and XTM data acquired in a previous study[39] are available under https://doi.org/10.3929/ethz-b-000505938.

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

## Acknowledgements

S.M., C.S., N.W., R.S., and V.W. acknowledge financial support from a European Research Council Starting Grant (6800070). This research used resources of the Advanced Photon Source, a U.S. Department of Energy (DOE) Office of Science User Facility operated for the DOE Office of Science by Argonne National Laboratory under Contract No. DE-AC02-06CH11357.

## Author contributions

S.M. and C.S. performed the electrochemical cycling experiments. S.M. and N.W. prepared the samples for the imaging campaign. F.D.C. and V.D.A. prepared the 32-ID-C beamline of the Advanced Photon Source at the Argonne National Laboratory for the XTM measurements. S.M., C.S, N.W., R.S and V.D.A. conducted the XTM experiments. C.S. reconstructed the imaging data. S.M. generated the artificial learning data including synthetic structures and x-ray simulation data. S.M. performed the segmentation of the image data. S.M. and C.S. analyzed the microstructure data. S.M., C.S., E.K. and V.W. wrote the manuscript. All authors contributed to scientific discussions.

## Competing interests

The authors declare no competing interests.
