## [Peer Review File · Nature Communications]

Reviewers' comments:

Reviewer #1 (Remarks to the Author):

Dear authors, please find below my comments.

- Line 24, Abstract.

Would the term "Parameter-free" be more adequate than "Model-free"? Model is a term with a wide application/vague definition.

- Line 39.

Low effective transport increases the ionic resistance, leading to voltage losses (overpotentials) and a smaller usable capacity [5]. Poor ionic transport properties are particularly detrimental for fast charge, required for automotive application, as it drastically reduces the rate capability [*].

[*] Andrew M. Colclasure, Alison R. Dunlop, Stephen E. Trask, Bryant J. Polzin, Andrew N. Jansen, and Kandler Smith, Requirements for Enabling Extreme Fast Charging of High Energy Density Li-Ion Cells while Avoiding Lithium Plating, *Journal of The Electrochemical Society*, 166 (8) A1412-A1424 (2019).

DOI: 10.1149/2.0451908jes

E.g. figure 12 of the article.

- Line 60.

Please mitigate this statement, as suggested below. While it is true segmenting graphite is more challenging than other materials (e.g. NMC), it is doable with a decent match with other techniques (see ref 23). The "large segmentation error" is otherwise definitely true if attributed for the additive phase.

[...] which make up a large fraction of commercial negative electrodes do not offer such contrast, potentially leading to larger segmentation errors compared with other electrode materials such as NMC [16].

- Line 64

Identification of the binder domain, which also typically contains conductive additives such as nanoscale carbon black in case of low-conductivity active particles, [17], [18] requires high contrast due to the low atomic number of carbon and high spatial resolution imaging, which results in a small field of view leading to a statistically insignificant number of active particles being imaged. Indeed, the order of magnitude difference between the nanoscale features of the additive phase (additives nanopore size range from 5 to 150 nm [*]) and the active material particle size (NMC cathode and graphite electrode particle size is typically in the 1-10 μm range [**]) implies that imagings performed on a field of view representative of the active material spatial distribution is very unlikely to have the required image resolution to being able to distinguish the additives' features, resulting in an additional uncertainty for the electrode microstructure properties [23].

[*] L. Zielke, T. Hutzenlau, D. R. Wheeler, C.-Wei Chao, I. Manke, A. Hilger, N. Paust, R. Zengerle, and S. Thiele, Three-Phase Multiscale Modeling of a LiCoO₂ Cathode: Combining the Advantages of FIB-SEM Imaging and X-Ray Tomography, *Adv. Energy Mater.* 2015, 5, 1401612

[**] F. L. E. Usseglio-Viretta, D. P. Finegan, A. Colclasure, T. M. M. Heenan, D. Abraham, P. Shearing, and K. Smith, Quantitative Relationships Between Pore Tortuosity, Pore Topology, and Solid Particle Morphology Using a Novel Discrete Particle Size Algorithm, *Quantitative Relationships Between Pore Tortuosity, Pore Topology, and Solid Particle Morphology Using a Novel Discrete Particle Size Algorithm*

- Line 66

In this work, we show how supervised / semi-supervised / unsupervised (please add adjective that

corresponds) deep learning algorithms can be used to enable semantic segmentation of [...]

- Line 70

Literature review on machine learning-based segmentation applied on electrode battery materials should have its own paragraph given the topic of the article. What is the current state/limitations of machine learning-based segmentation for electrode battery? It would be interesting to mention that machine learning algorithms not only have the potential to provide a better-quality segmentation, but also to provide additional information out of reach of classic threshold-based methods (e.g. identifying specific features such as cracks). For example, crack detection in lithium-ion cells using machine learning (<https://doi.org/10.1016/j.commatsci.2017.05.012>) and Machine-learning-revealed statistics of the particle-carbon/binder detachment in lithium-ion battery cathodes (<https://doi.org/10.1038/s41467-020-16233-5>) should be quoted.

- Line 75

One key challenge in employing deep learning techniques is having sufficient and quality learning data for the algorithm so that it can operate fully automated on the dataset of interest.

Please elaborate on the required amount of data required to train machine learning algorithm (here or later in the text if more relevant). Is there enough data within a unique 3D representative volume with hundreds-thousands of particles? Say otherwise, could you provide an estimation, or provide an educated guess on the required number of particles to train the data set? The rule of thumb is 1000 representative images are required to train for each class – but that's questionable and may not be relevant here. As electrodes exhibit different heterogeneities (shape, cracks, etc.), this number is expected to vary a lot. In addition, is there a need to combine imaging performed at different scale (let's say, one at a very fine image resolution to train the algorithm on the additives, and another with a coarser image resolution to train the algorithm on the active material particles) ?

- Line 77

Typically, learning data for supervised training is real data that has been successfully segmented and labeled; however, since such data is experimentally difficult and time consuming to obtain, we present a systematic approach to computationally generate learning data for LIB electrodes to enhance the quality of the learning-based segmentation as quantified by the Dice score.

Interesting approach, although, is there a risk to over train the algorithm on recognizing features that are present in the generated data, and under train the algorithm on features not present in the generated data but present in the real microstructure, thus introducing a bias in the method? One naïve example (not necessarily applicable to your case, but for the sake of illustrating my point) would be to train an algorithm on recognizing cracks, using generated particles for which cracks are always open-porosity cracks, and then use the algorithm on actual data set for which particles have both open porosity and close porosity cracks, resulting in the algorithm correctly identifying the open porosity cracks but missing the close porosity cracks.

- Line 103.

We image three samples of each because the sample size of below 70 μm is close to the limit of representativeness of the heterogeneous material, with even commercial electrodes exhibiting heterogeneity at this length scale [5].

Have you verified after segmentation the heterogeneity/RVE, at least on volume fractions ? While it is not the focus of the analysis, it is worth mentioning.

- Line 114

Specifically, it is shown that the neural network model trained only on real datasets is outperformed by a network trained hybrid learning datasets (Figure 1c).

Could you elaborate on this statement?

- What features/phases are less correctly identified when using only the real data sets.
- Why such difference, what is your explanation? Are some features in the real data set incorrectly labelled/confused, are the artificial learning data sets tuned to train the algorithm on some specific features to correct this? Or is this simply a lack of data?
- If this is a lack of data, does it imply the field of view is too small? A field of view may be large enough to be qualified as an RVE, but still too small to train an algorithm. Estimating the RVE for porosity, particle size or tortuosity would be interesting for your analysis as you could provide an estimation of the cumulated volume size required to train the algorithm as a multiple of the RVE. For instance, if the maximum RVE calculated on various microstructure parameters (RVE is property-dependent) is $100 \times 100 \times 100 \text{ } \mu\text{m}^3$, and that you need to use 50 volumes of size $70 \times 70 \times 70 \text{ } \mu\text{m}^3$ to train correctly the algorithm (I used arbitrary numbers), you could then conclude the required amount of data required to train machine learning algorithm is ~ 17 times the RVE size.

- Line 119.

As depicted in Figure 2b, simple thresholding (here done with a k-means implementation) does not nearly lead to satisfactory results.

How look likes segmentation when the threshold is done manually, for each phase? Complex images can, sometimes, be segmented even with simple methods such as threshold/edge detection when done sequentially, with a final step to combines the different results. Although, I agree that achieving a decent segmentation based on thresholding, if even possible, would take a significant amount of times and tuning, and is likely to be less generic than the method presented in your work. I realize that may be what you meant saying 'simple thresholding'...

I saw you provide manual segmentation in the supplemental information (5.), you may comment them in the manuscript.

- Line 129

However, that fact that the human eye can pick out different phases despite the unfavorable gray value histogram (Figure 2d) suggests that a learning-based approach to semantic segmentation would be applicable to the datasets.

If I would ask my colleagues to do a blind-segmentation of fig 2a, only specifying the different phases and their overall specificities (e.g., graphite are near spherical particles, additives are smaller particles with nano porosity), I would not bet they achieve the same segmentation between them. Did you perform a blind manual segmentation to support your statement? The method may be simplistic or complex, if the raw image quality is so poor, the information may be lost – and the resulting segmentation, even if it looks good, may be biased (that's more a general comment, than a critic of your work).

- Line 157

The real learning data is used to train a neural network model. 1000 volume pairs (the 3D image data and label map) of dimensions $15.6 \times 15.6 \times 3 \text{ } \mu\text{m}$ ($256 \times 256 \times 48$ voxel) are randomly sampled from the learning data such that the smallest dimension is equally likely to be along the x, y, or z axis.

- Why are you using non-cubic domains, so elongated? Is there a reason to have an anisotropic domain?
- With such small third dimension, the likelihood to cut particles are high. Would this not add a bias,

training algorithm to identify graphite particles as most of the time being truncated?

- Why training on 1000 subvolumes, while not directly on the whole volume? This would avoid truncated particles.

- Line 166

Training and deploying the neural network on the real learning data shows good accuracy for 167 the silicon phase (Figure 3e and f) [...]

- Am I correct that what you called the ground truth is the segmentation result obtained from a previous work segmentation? If yes, this implies it is not, strictly speaking, a ground truth but instead a 'state of the art segmentation' or 'best previous segmentation' etc. The difference means that whatever is the (low) error of the previous segmentation, the algorithm is trained somewhat to reproduce in a systematic way the same error.

- Could the good accuracy for the silicon due to its much lower size compared with the graphite particles? I.e., in the training volume there are more silicon 'particles' to be trained compared with the graphite particles. Although, if true, the carbon-black binder should be also well identified, which is not the case. What reason could explain silicon is correctly identified, but not the additives?

- Line 173

[..] such that can the most likely segmentation failure modes are extensively trained [...]

What does mean 'segmentation failure modes'?

- Line 196

The resulting simulated tomography images (Figure 4f) together with the labeled digital twins form the artificial learning datasets

To better judge of the accurateness of the segmentation problem for these generated structures and associated tomography volumes, compared with the actual ones, I suggest adding a simplified version of figure 2, but applied to these generated tomography scans. Is the histogram still as convoluted? Is the threshold-based method still inaccurate? Significant differences would imply that the training is performed on an intrinsic different set of data. Maybe, with a closer to reality generated tomography, the required size of training data set could be reduced, and/or for a same training data set size, the predictions would be even better.

- Line 201

The procedure is kept the same as before, except that this time, of 1000 volume pairs, 500 are sampled from real learning datasets and 500 from artificial learning datasets, which is why we refer to it as hybrid learning data.

Training on only real dataset provides inaccurate predictions. What about training on only generated data set? Does it differ from the case where both real and generated data set were used for the training?

- Line 204

Comparing the results from the network trained on hybrid learning data (consisting of 50% of artificial datasets) to the network trained only on the real learning data (as discussed above), the benefits of adding artificial datasets to the learning data can be seen: the quality of the segmentation is shown visually in Figure 5 and assessed numerically with Dice coefficients [46] (Table 1).

- Figure 5d,f,h. The CBD phase seems not very well labelled. 5d seems to neglect the nano porosity, while 5f and 5h under represents the CBD phase – from visual comparison with 5b. Expected volume fraction from weight fractions are known (line 280) and could be used to try discriminate the segmentation, although it is not sure the field of view is representative of the volume fractions.

Considering the imaging resolution size may be still too coarse to resolve the CBD nanopore, would it be more reasonable to instead try to catch the micrometer scale representation (as seen in fig d)? The resulting medium could be considered as a heterogeneous medium at this scale, and thus assigned with effective properties (CBD nanoporosity has been quantified at <https://doi.org/10.1021/acsaem.8b00501>).

- Another issue is the lack of contact between the graphite, silicon, and CBD phases. Phases seem not to touch each other, which prevents estimation of active surface area. Did you calculate interfacial area to verify if contact issue is real or is only not well represented with a 2d slice view. Adding the interfacial area – even with a basic method such as counting the face between voxels belonging to different phases – would provide a relative estimation between the three cases.

- Providing a short definition of Dice coefficients could be instructive for non machine learning experts, and how to interpret them (e.g., a dimensional parameter, ranging from 0 to 1, the higher the better, a value of 1 means ***).

- Line 209

To evaluate and compare the different neural network models, 400 sequential slices of the TXTM dataset of one of the three pristine samples are manually segmented with the help of the Dragonfly software.

Did you compare with segmentations realized with the WEKA Fiji plugin? Being free and open source compared with Dragonfly, WEKA is often the first try for segmentation of microstructure too complex for basic thresholding.

- Line 250

This procedure improves the segmentation of the internal structure carbon black-binder domain (Figure 5h) [...]

This is debatable, see previous comments.

- Line 262.

The cylindrical tomography images are binned by a factor of 2.22 resulting in a voxel size of 61 nm.

Additives nanopore size range from 5 to 150 nm [*]. You could include this information as it provides some number to explain why the CBD small features are difficult to segment.

[*] L. Zielke, T. Hutzenlau, D. R. Wheeler, C.-Wei Chao, I. Manke, A. Hilger, N. Paust, R. Zengerle, and S. Thiele, Three-Phase Multiscale Modeling of a LiCoO₂ Cathode: Combining the Advantages of FIB–SEM Imaging and X-Ray Tomography, *Adv. Energy Mater.* 2015, 5, 1401612

- 282 Microstructural Analysis and Changes with Cycling

The evolution analysis is limited by the small field of view (representativity issue), thus establishing trends is risky, as, as indicated in the text “we are not tracking the microstructural evolution of the same sub-section of electrode with cycling”. There is a real possibility of over-interpreting microstructure parameter evolution with cycling in this section. Then “we therefore focus on robust

trends across the three samples and detailed analysis of changes around specific phases” seems a too strong statement, especially considering only 3 small samples are considered.

- Line 296

The exact number may be a slight underestimate since the identification of graphite particles during segmentation is based on edge-enhancement of the tomography data (see Figure 2a), making it difficult to distinguish bright graphite edges and the bright carbon-black and binder domain signal.

“may be a strong underestimate” seems more realistic.

- Line 313

We perform numerical diffusion simulations on the electrode structures and find that the effective transport coefficient is 0.14.

Please explain how you realized the calculation briefly (Tau factor?) and the choice of the boundary conditions. Volume being small, the calculation may suffer from edge effects (isolated pores located at the domain edges, named “unknown cluster”, bias the calculation for low volumes). As well, for low volumes, the choice of the boundary conditions will impact the solution. A possible way to increase the level of confidence in the result would be calculate the effective transport coefficient with Dirichlet-Dirichlet boundary conditions and then again but with Neumann-Neumann (with an additional fixed point) to verify the two bounds are not far from each other. See J. Joos et al., *J. Power Sources*, 246, 819 (2014) for the concept of unknown clusters, and F. Usseglio-Viretta et al., *ECS Transactions*, 77 (11) 1095-1118 (2017) for the impact of boundary conditions.

Given the uncertainty of the CBD segmentation (cf. Figure 5), the diffusion coefficient should be calculated with the CBD phase obtained with figures 5d and 5h to provide uncertainty bounds. For both cases, the graphite segmented with the hybrid learning data + thresholding would be used. The same approach could also be used for the interfacial area calculations.

Of interest also would be to calculate the diffusion coefficient when CBD and/or silicon are not represented to have an estimation of the error performed when only graphite and pore are considered, as they are the phases the most simple to segment. This is what you did (line 320), although what is the evolution of the exponential coefficient alpha ($\text{tortuosity} = \text{porosity}(1 - \alpha)$) between the two cases? Alpha is likely to increase when the CBD is added to the analysis, indicating that the CBD impact on tortuosity is more than just a reduction of porosity. Such result is important as it indicates one cannot simply extrapolate a tortuosity-porosity relationship established on a graphite pore domain to a graphite CBD pore domain, as previously investigated in [23].

- Line 330

Nonetheless, quantification of the morphology of the carbon black-binder domain is one step towards understanding ionic and electronic performance of the electrode and it can be used to form the basis of computer-generated carbon black-binder domains [34][54][55][56][*]

[*] A. N. Mistry, K. Smith, and P. P. Mukherjee, *Secondary-Phase Stochastics in Lithium-Ion Battery Electrodes*, *ACS Appl. Mater. Interfaces* 2018, 10, 7, 6317–6326

- Line 365

This means that trends induced by the cycling behavior of the silicon can easily be hidden within the statistical variation found sample to sample, and would require a larger field of view to reach the representativity scale of the silicon spatial distribution.

- Line 381

The increase in the amount of carbon black-binder domain in subsequent volume shells (Figure 8d) can be explained by the carbon black-binder domain forming clusters in the pore space between active particles.

Very interesting result that should be stressed more as it can guide development of CBD generating algorithms. Some CBD generation algorithm such as the one from Hein [56] and Usseglio [*] are using a 'bridge' approach, assuming CBD is likely to be between neighbored particles, which could fit with this result, as CBD must be present in the pore space between particles. Other algorithm, that tends to generate CBD preferentially at the particle surface (thus creating a thin layer at the surface with more or less surface rugosity) would less fit this result.

[* no need to be quoted, but FYI] MATBOX: Microstructure Analysis Toolbox,
https://github.com/NREL/MATBOX_Microstructure_analysis_toolbox

- Line 386

Here the opposite trend is found, indicating CBD generation algorithms that favor deposition at the particle surface are more suited for silicon, compared with CBD generation algorithms built around a 'bridge' approach.

Results suggests combining different CBD generation algorithms/methods may be required to adequality model CBD spatial distribution in graphite silicon electrodes.

- Supplemental information

7. Additional Structural Analysis

How are diameters calculated? There is a whole family of numerical methods to calculate a diameter. Knowing the method help better judge the results.

Reviewer #2 (Remarks to the Author):

This manuscript pertains to the application of deep learning techniques, namely an adaptation of the 3D-UNet convolutional neural network for segmentation of four-phase battery electrodes consisting of graphite, silicon, CBD, and void. Yet beyond the application of DL for segmentation, the paper is incredibly broad, representing the manufacturing, cycling, and imaging of electrodes, plus the study of a variety of segmentation techniques (both manual and twists on the traditional DL processes). This manuscript represents the compilation of a significant amount of work, but with the focus on the very important topic of image segmentation. In particular, this work represents the first segmentation of a battery electrode that includes image-based representation of the CBD at a scale and resolution that is usable for 3D mesoscale simulation. While the methods that are highlighted in the manuscript are worthy on their own, the image datasets that are promised for publication will be widely used by many other researchers. As such, I feel that it warrants publication in Nature Communications, after a few minor modifications are made and questions answered, which are listed below.

1. Line 73. You state that DL has not been applied to battery electrodes. Yet there have been two recent publications this area, one in this journal (doi: 10.1038/s41467-020-16233-5) and another preprint from last year (arxiv: 1910.10793), both of which apply CNNs to cathode and anode images, respectively.

2. To obtain segmentations for a large domain, you create smaller subdomains and "average" their results in the overlapped region using a radial weighting function. Do you also do this during training, or just at inference time? Using this overlap during training could be particularly important because of the small amount of data that can fit in CPU memory; you specifically call out that some graphite particles can span the training subdomains (line 235).

3. Your “basic structure” generation algorithm seems rather ad-hoc, especially given how researchers have recently used physics-based simulations to generate microstructures (see doi: 10.1016/j.jpowsour.2019.227285 and doi: 10.1021/acsami.0c08251 for examples). Now, it may not matter that much because of your application of CycleGAN afterwards. But you should better justify your present approach and at least acknowledge the increased fidelity of simulated mesostructures in the literature.
4. Regarding the CycleGAN, there was a nice paper recently published in this journal family (doi: 10.1038/s41524-020-0340-7) that pioneered using CycleGANs in the way that you describe, that should be cited here (around line 184).
5. Line 199: Why are your artificial learning datasets so much smaller in size than the image-based datasets? Do you think this influences the quality of the model?
6. Given the difficulty in creating manual segmentations for the domains, how good do you think your Dice score comparisons are? Could you instead compare to the “hybrid” data, both the manually segmented and generated images?
7. It appears that all of your quantitative metrics for image segmentation quality are on the training domain and none on held-out samples. It is typical in the ML community to not only compare model accuracy in the training domain, but also in held-out samples that have also been manually segmented. It is common for models to perform very well on the training domain, but less-so on new, held-out data.
8. Line 333: Consider reference the work of Franco and coworkers, specifically doi: 10.1016/j.jpowsour.2019.227285.
9. Line 353: You say that the CBD does not deform extensively. Do you mean it does not plastically/inelastically deform? I suspect that during a charge-discharge cycle, it does deform more significantly.
10. You use the phrase “ground truth” to denote the manually segmented image data used for training. Typically, however, “ground truth” refers to training (or validation) data that is known to be exactly correct. Your manual segmentations, while likely good for training, are not perfectly true. You should use “training data” rather than “ground truth”.
11. SI Section 3.1: The porosity of 5-40% of CBD seems rather low compared to doi: 10.1002/aenm.201401612.
12. SI Section 6: You refer to the output of the network as a “probability map.” You do not have enough details of the network here to verify, but typically these networks output a softmax layer, which is not to be exactly interpreted as a probability map.

Reviewer #3 (Remarks to the Author):

This work used deep Learning-based approach to analyze the X-ray tomographic data, and applied the technique for Li-ion batteries. As the authors stated, high-quality images are either technically not possible or not affordable from a time or resource perspective. In such situations, deep learning-based method can be very helpful for accurate segmentation of volumetric datasets. The authors showed the morphological evolution of graphite-silicon anodes, particularly carbon black-binder domain detached at regions close to the silicon particles. These are helpful to understand the impact on transport.

While the key part of this work is about machine learning, the machine learning approach itself was not new (borrowed a specific implementation of the 3D-Unet architecture from Baumgartner et al.). The approach has been well applied for medical imaging already. Since this work is machine learning focused while the work itself did not provide sufficient advances on machine learning, it would fit better a more applied journal.

In addition, the authors have previous published papers such as "Multimodal nanoscale tomographic imaging for battery electrodes", which shows a more comprehensive study of graphite-silicon composite electrodes based on tomographic imaging. Thus, results on tomographic imaging of silicon particles do not represent major advances.

On the technical side, the following is suggested.

1. No mathematical analysis was provided on the machine learning approach. Such analysis should be discussed.
2. Rigorous quantification of the machine learning performance should be provided.
3. A comparison to other benchmark techniques should be provided.

Reviewer #1 (Remarks to the Author):

Dear authors, please find below my comments.

We thank the reviewer for their detailed feedback on the manuscript and the helpful suggestions. We have revised the main text and the supplementary information (SI) according to the input and present the point-by-point reply to the comments, questions, and suggestions below.

- Line 24, Abstract.

Would the term ‘Parameter-free’ be more adequate than ‘Model-free’? Model is a term with a wide application/vague definition.

We thank the reviewer for pointing out a source for possible misunderstanding. We intended to highlight that no physical model is behind the machine-learning algorithm. However, parameter-free could also lead to confusion as a neural network has many free parameters and hyper-parameters. Therefore, we dispense with further specifications in the abstract and simply state :

“Here, we demonstrate a methodology for using deep-learning tools to achieve reliable segmentations of volumetric images of electrodes on which standard segmentation approaches fail due to insufficient contrast. We implement the 3D U-Net architecture for segmentation, and, to overcome the limitations of training data obtained experimentally through imaging, we show how synthetic learning data, consisting of digital twins of our electrodes and their tomographic reconstructions, can be generated and used to enhance network performance.”

- Line 39.

Low effective transport increases the ionic resistance, leading to voltage losses (overpotentials) and a smaller usable capacity [5]. Poor ionic transport properties are particularly detrimental for fast charge, required for automotive application, as it drastically reduces the rate capability [*].

[*] Andrew M. Colclasure, Alison R. Dunlop, Stephen E. Trask, Bryant J. Polzin, Andrew N. Jansen, and Kandler Smith, Requirements for Enabling Extreme Fast Charging of High Energy Density Li-Ion Cells while Avoiding Lithium Plating, Journal of The Electrochemical Society, 166 (8) A1412-A1424 (2019).

DOI: 10.1149/2.0451908jes

E.g. figure 12 of the article.

Thanks for this suggestion. We rephrased the sentence and added the citation (Reference [6]) to highlight the importance of ionic transport properties on fast charging. It now reads (Line 30):

“Low effective transport increases the ionic resistance, leading to voltage losses (overpotentials), a smaller usable capacity [5] and a reduced rate capability, particularly during fast charge required for automotive applications [6].”

- Line 60.

Please mitigate this statement, as suggested below. While it is true segmenting graphite is more challenging than other materials (e.g. NMC), it is doable with a decent match with other techniques (see ref 23). The “large segmentation error” is otherwise definitely true if attributed for the additive phase.

[...] which make up a large fraction of commercial negative electrodes do not offer such contrast, potentially leading to larger segmentation errors compared with other electrode materials such as NMC [16].

We rephrased as suggested by the reviewer (Line 47):

“However, obtaining 3D reconstructions that can be accurately segmented and quantitatively analyzed is still a challenge, primarily due to (i) the diverging length scales present in LIB electrodes, (ii) the low contrast between key components, and (iii) the low attenuation of carbon-based materials [16]. Active materials such as $\text{Li}(\text{Ni},\text{Mn},\text{Co})\text{O}_2$ have particle dimensions in the range of 1-10 μm [17] and contain transition metal elements that provide good contrast during absorption-based imaging and hence a reliable identification of the particles. Graphitic active particles, which make up a large fraction of commercial negative electrodes, do not offer such contrast potentially leading to large errors in segmentation [18].”

- Line 64

Identification of the binder domain, which also typically contains conductive additives such as nanoscale carbon black in case of low-conductivity active particles, [17], [18] requires high contrast due to the low atomic number of carbon and high spatial resolution imaging, which results in a small field of view leading to a statistically insignificant number of active particles being imaged. Indeed, the order of magnitude difference between the nanoscale features of the additive phase (additives nanopore size range from 5 to 150 nm [*]) and the active material particle size (NMC cathode and graphite electrode particle size is typically in the 1-10 μm range [**]) implies that imaging performed on a field of view representative of the active material spatial distribution is very unlikely to have the required image resolution to being able to distinguish the additives' features, resulting in an additional uncertainty for the electrode microstructure properties [23].

[*] L. Zielke, T. Hutzenlau, D. R. Wheeler, C.-Wei Chao, I. Manke, A. Hilger, N. Paust, R. Zengerle, and S. Thiele, Three-Phase Multiscale Modeling of a LiCoO_2 Cathode: Combining the Advantages of FIB-SEM Imaging and X-Ray Tomography, *Adv. Energy Mater.* 2015, 5, 1401612

[**] F. L. E. Usseglio-Viretta, D. P. Finegan, A. Colclasure, T. M. M. Heenan, D. Abraham, P. Shearing, and K. Smith, Quantitative Relationships Between Pore Tortuosity, Pore Topology, and Solid Particle Morphology Using a Novel Discrete Particle Size Algorithm, *Quantitative Relationships Between Pore Tortuosity, Pore Topology, and Solid Particle Morphology Using a Novel Discrete Particle Size Algorithm*

We rephrased the relevant paragraph in order to clarify this and we added the suggested citations (References [17],[19]). The new version now reads (Line 50):

“Active materials such as $\text{Li}(\text{Ni},\text{Mn},\text{Co})\text{O}_2$ have particle dimensions in the range of 1-10 μm [17] and contain transition metal elements that provide good contrast during absorption-based imaging and hence a reliable identification of the particles. [...]

Furthermore, identification of the polymeric binder domain, which, in case of low-conductivity active particles also contains conductive additives such as nanoscale carbon black, requires not only high contrast (due to the low atomic number of carbon) but also high spatial resolution imaging (due to the nanometer sized pores and structural features ranging from 5 to 150 nm [19], [20], [21]). Techniques with high resolution ($\sim 10\text{-}50$ nm voxel edge lengths) have limited fields of view ($\sim (10 - 50 \mu\text{m})^3$) and consequently only a small sample volume with a limited number of active particles can be imaged quantitatively, which is problematic since analyzing the active material distribution requires a field of view in the range of 5x the largest particle size or $\sim 100 \mu\text{m}$ to centimeters [5] in order to be representative. Furthermore, high-resolution imaging typically requires long imaging times, often making it prohibitive to obtain statistically relevant data on the electrode scale by imaging many small samples serially [22], [23].”

- Line 66

In this work, we show how supervised / semi-supervised / unsupervised (please add adjective that corresponds) deep learning algorithms can be used to enable semantic segmentation of [...]

We added the word “supervised” as suggested.

- Line 70

Literature review on machine learning-based segmentation applied on electrode battery materials should have its own paragraph given the topic of the article. What is the current state/limitations of machine learning-based segmentation for electrode battery? It would be interesting to mention that machine learning algorithms not only have the potential to provide a better-quality segmentation, but also to provide additional information out of reach of classic threshold-bases methods (e.g. identifying specific features such as cracks). For example, crack detection in lithium-ion cells using machine learning (<https://doi.org/10.1016/j.commatsci.2017.05.012>) and Machine-learning-revealed statistics of the particle-carbon/binder detachment in lithium-ion battery cathodes (<https://doi.org/10.1038/s41467-020-16233-5>) should be quoted.

We thank the reviewer for this idea and complemented the literature review with the mentioned publication (References [31],32]). The literature review now reads:

“In this work, we show how supervised, deep learning can help address the challenges associated with semantic segmentation of high-resolution, volumetric image data of LIB electrodes. While deep learning algorithms have been applied to assess the state of health of a battery [24], to improve the microstructural design [25], [26], or to detect defects during manufacturing [27], microstructure segmentation continues to rely heavily on adapted filtering followed by simple thresholding operations [28]. Even though algorithms originating in the field of machine learning are beginning to be used for segmentation [29], [30], crack detection [31] and particle detachment [32] to date, deep learning-based approaches are mainly used on high-contrast systems (i.e. cathodes), and algorithms for complete 3D segmentation such as those which are used in medicine to analyze tomographic full-body scans and identify organs [33], have not been applied to semantic segmentation of LIB electrode datasets. Here, we work with the 3D U-Net architecture for semantic segmentation of volumetric image data [34], and show how it can be trained and implemented to segment 3D reconstruction of electrodes into all their different material components (i.e, active particle, binder, pore). “

- Line 75

One key challenge in employing deep learning techniques is having sufficient and quality learning data for the algorithm so that it can operate fully automated on the dataset of interest.

Please elaborate on the required amount of data required to train machine learning algorithm (here or later in the text if more relevant). Is there enough data within a unique 3D representative volume with hundreds-thousands of particles? Say otherwise, could you provide an estimation, or provide an educated guess on the required number of particles to train the data set? The rule of thumb is 1000 representative images are required to train for each class – but that’s questionable and may not be relevant here. As electrodes exhibit different heterogeneities (shape, cracks, etc.), this number is expected to vary a lot. In addition, is there a need to combine imaging performed at different scale (let’s say, one at a very fine image resolution to train the algorithm on the additives, and another with a coarser image resolution to train the algorithm on the active material particles)

The reviewer raises a very important and interesting question. There is no rule of thumb for the required amount of data as it depends strongly on the specifics of the application, the inter and intra class variance, the image quality, etc. We use synthetic data to account for the heterogeneity found in samples that may not be correctly accounted for in real data. There could indeed be other approaches to this problem; however, ultimately the training data should resemble the data to be analyzed.

We now explicitly discuss this throughout our manuscript, so the reader is alerted to the complexity of considerations to be made.

In the introduction (Line 88), we write:

“A key challenge in employing a convolutional neural network like 3D U-Net for segmentation of volumetric image data is having sufficient and high quality learning data so that the algorithm can operate fully automated on the dataset of interest. Learning data consists of image pairs: the “input” image and the “output” image, which is the successfully segmented and labeled version of the “input” image. Typically, learning data is real, experimental data that has been segmented and labeled. However, as explained above, no single x-ray imaging technique provides the contrast, resolution, and field-of-view in 3D needed to accurately segment all material phases in an electrode. Multimodal imaging approaches (e.g., combining x-ray absorption contrast tomography with ptychographic x-ray computed tomography [38]) makes it possible to achieve segmented and labeled datasets; however, such experiments are difficult and time-consuming. Indeed, we show that the amount of high quality multimodal image data obtainable during typical experimental beamtimes at a shared synchrotron facility is insufficient for training of the neural network and leads to only partial segmentation accuracy, as quantified by the Dice score [39].”

Later in the main text (line 204), we write:

*“While the learning-based segmentation is a clear improvement over simple thresholding or random-walk based segmentation approaches (**Figure 2**), we attribute the less than satisfactory segmentation to the limitations of the real learning datasets. For example, in the real dataset, large graphite particles may be cut in the sample preparation prior to the XTM and PXCT imaging, which means that the neural network will tend to underestimate particle sizes and thereby misattribute the portions of graphite particles to pore space.*

Simply adding to learning data by taking more high-resolution images is impractical from a time and resource perspective. Each high-resolution PXCT scan takes 50 hours. Furthermore, we have recognized that the limited sample volumes leads to challenges in terms of representative nature of the real learning data set.

For this reason, we turn to computationally generated (i.e., “synthetic” or “artificial”) learning data, which offers control over the data content (i.e., particle sizes and volume fractions) such that the most common segmentation failures (e.g., misinterpreting pore space for graphite) are extensively trained and the neural network is more robust...”

- Line 77

Typically, learning data for supervised training is real data that has been successfully segmented and labeled; however, since such data is experimentally difficult and time consuming to obtain, we present a systematic approach to computationally generate learning data for LIB electrodes to enhance the quality of the learning-based segmentation as quantified by the Dice score.

Interesting approach, although, is there a risk to over train the algorithm on recognizing features that are present in the generated data, and under train the algorithm on features not present in the generated data but present in the real microstructure, thus introducing a bias in the method? One naïve example (not necessarily applicable to your case, but for the sake of illustrating my point) would be to train an algorithm on recognizing cracks, using generated particles for which cracks are always open-porosity cracks, and then use the algorithm on actual data set for which particles have both open porosity and close porosity cracks, resulting in the algorithm correctly identifying the open porosity cracks but missing the close porosity cracks.

We thank the reviewer for bringing this up this point since introducing a bias towards synthetic data is an issue, especially since the synthetic data is not perfect.

To avoid overtraining the algorithm on synthetic data, we use a mixture of real and artificial learning data (we call it hybrid data).

We explain how the hybrid learning data is made up in the section “Deep Learning Segmentation with Synthetic and Real Learning Data”.

“Using 21 artificial learning datasets (20 μm x 20 μm x 20 μm in size) in addition to the two real datasets (cylinders of 40 μm diameter and 20 μm height), the neural network models are again trained. The procedure for training is kept the same as before, except that, this time, 500 volume pairs are sampled from real learning datasets and 500 volume pairs from artificial learning datasets. We refer to this as hybrid learning data.”

- Line 103.

We image three samples of each because the sample size of below 70 μm is close to the limit of representativeness of the heterogeneous material, with even commercial electrodes exhibiting heterogeneity at this length scale [5].

Have you verified after segmentation the heterogeneity/RVE, at least on volume fractions? While it is not the focus of the analysis, it is worth mentioning.

Yes, we did do this. Please see Figure 7a.

Sample heterogeneity and a verification of the representative sample size are important although sometimes hard to answer questions when analyzing microstructures. For each sample, we analyzed the volume fraction of each phase composition of each sample as well as the average for the three samples of a kind as can be seen in Figure 7a in the manuscript. The reference line (in red) indicates the average volume fraction expected for each phase based on the electrode manufacturing recipe. The volume fractions for pore space, graphite and CBD are close to the reference value. If we assume a relatively accurate segmentation, this indicates that our sample size is in the range of the representative volume element for these three phases. Due to the low silicon volume fraction (around 6%), the statistical variations from sample to sample are higher and would require a larger field of view to reach the representability scale of the silicon spatial distribution.

- Line 114

Specifically, it is shown that the neural network model trained only on real datasets is outperformed by a network trained hybrid learning datasets (Figure 1c).

Could you elaborate on this statement?

- What features/phases are less correctly identified when using only the real datasets.

- Why such difference, what is your explanation? Are some features in the real data set incorrectly labelled/confused, are the artificial learning data sets tuned to train the algorithm on some specific features to correct this? Or is this simply a lack of data?.

- If this is a lack of data, does it imply the field of view is too small? A field of view may be large enough to be qualified as an RVE, but still too small to train an algorithm. Estimating the RVE for porosity, particle size or tortuosity would be interesting for your analysis as you could provide an estimation of the cumulated volume size required to train the algorithm as a multiple of the RVE. For instance, if the maximum RVE calculated on various microstructure parameters (RVE is property-dependent) is 100 x 100 x 100 μm^3 , and that you need to use 50 volumes of size 70 x 70 x 70 μm^3 to train correctly the algorithm (I used arbitrary numbers), you could then conclude the required amount of data required to train machine learning algorithm is ~ 17 times the RVE size.

- Line 119.

As depicted in Figure 2b, simple thresholding (here done with a k-means implementation) does not nearly lead to satisfactory results.

How look likes segmentation when the threshold is done manually, for each phase? Complex images can, sometimes, be segmented even with simple methods such as threshold/edge detection when done sequentially, with a final step to combines the different results. Although, I agree that achieving a decent segmentation based on thresholding, if even possible, would take a significant amount of times and tuning, and is likely to be less generic than the method presented in your work. I realize that may be what you meant saying 'simple thresholding'...I saw you provide manual segmentation in the supplemental information (5.), you may comment them in the manuscript.

During the experimental processes leading to this manuscript, we followed numerous segmentation approaches (thresholding, random walk, active contours, level set, region growth, graph cuts). Apart from the fact that no combination of these methods led to usable segmentations, they also needed a large amount of tuning to the respective datasets and were, as you mentioned, not generic.

We added a reference to Section 5 Manual Segmentation in the Supporting Information (Line 145):

"The need for advanced segmentation is clear. A cross-section of an example tomogram is shown in Figure 2a. Despite the low contrast, leading to an unfavorable gray value histogram (Figure 2b), manual segmentation is possible (Supplementary Note 5), this is not feasible for rapid analysis of the twelve imaged volumes. As depicted in Figure 2c, conventional methods like simple thresholding (here done with a k-means implementation) do not lead to satisfactory results."

- Line 129

However, that fact that the human eye can pick out different phases despite the unfavorable gray value histogram (Figure 2d) suggests that a learning-based approach to semantic segmentation would be applicable to the datasets.

If I would ask my colleagues to do a blind-segmentation of fig 2a, only specifying the different phases and their overall specificities (e.g., graphite are near spherical particles, additives are smaller particles with nano porosity), I would not bet they achieve the same segmentation between them. Did you perform a blind manual segmentation to support your statement? The method may be simplistic or complex, if the raw image quality is so poor, the information may be lost – and the resulting segmentation, even if it looks good, may be biased (that's more a general comment, than a critic of your work).

No multiple blind-segmentation was performed. On small excerpts, we generally find good accordance for silicon particles already among people that are not trained on interpreting tomography images. Also, graphite particles can be detected systematically by different colleagues. However, as this relies heavily on the edge information, it requires some experience to manually segment the graphite. For the CBD, the resolution is too low to detect single particles, which is why we rather segmented clusters without internal porosity in the manual segmentation.

- Line 157

The real learning data is used to train a neural network model. 1000 volume pairs (the 3D image data and label map) of dimensions 15.6x15.6x3 μm (256x256x48 voxel) are randomly sampled from the learning data such that the smallest dimension is equally likely to be along the x, y, or z axis.

- Why are you using non-cubic domains, so elongated? Is there a reason to have an anisotropic domain?

It would be ideal to train on the whole volume, however, memory limitations force us to work on smaller subvolumes. By choosing anisotropic shapes we only cut particles in one direction (and not in all three) and are able to compensate for this by having three sampling directions.

- With such small third dimension, the likelihood to cut particles are high. Would this not add a bias, training algorithm to identify graphite particles as most of the time being truncated?

Yes, we try to clarify this by writing in the main text (line 255):

“... artificial datasets are designed to be representative of the volume percents of each phase and the particle size distributions and contain more large graphite particles than the real datasets, which are often randomly cut in the sample preparation prior to imaging. Nonetheless, segmentation of the graphite and pore phases remains a challenge because of the low contrast between the graphite and pore space, the lack of distinguishing features within bulk graphite, and imaging artefacts around the edges of small graphite particles. Furthermore, the size of the receptive field [54] (i.e. the subset of the input image that influences the prediction for a certain voxel), is at times smaller than a graphite particle.”

- Why training on 1000 subvolumes, while not directly on the whole volume? This would avoid truncated particles.

Unfortunately, we are limited by the memory of the GPU. Otherwise, it would of course be preferable to directly train and evaluate the whole volume. We adjusted the manuscript that now reads (Line 177):

“Due to memory limitations, 1000 subvolume pairs (the 3D image data and label map) of dimensions 15.6x15.6x3 μm (256x256x48 voxel) are randomly sampled from the learning data such that the smallest dimension is equally likely to be along the x, y, or z axis. 50% of the training data is subjected to data augmentation based on basic image manipulations in order to render the neural network more robust and increase its ability to generalize while decreasing the potential danger of overfitting [51]. Specifically, we apply Gaussian noise and blurring, vary contrast and brightness, and use affine transformations. The model is trained for 100 epochs, which amounts to a total training time of 72 hours on a Nvidia TITAN Xp GPU.”

• Line 166

Training and deploying the neural network on the real learning data shows good accuracy for the silicon phase (Figure 3e and f) [...]

- Am I correct that what you called the ground truth is the segmentation result obtained from a previous work segmentation? If yes, this implies it is not, strictly speaking, a ground truth but instead a ‘state of the art segmentation’ or ‘best previous segmentation’ etc. The difference means that whatever is the (low) error of the previous segmentation, the algorithm is trained somewhat to reproduce in a systematic way the same error.

The “ground truth” is a manually segmented structure. We have revised Figure 1 to better show how different real and synthetic datasets are used.

- Could the good accuracy for the silicon due to its much lower size compared with the graphite particles? I.e., in the training volume there are more silicon ‘particles’ to be trained compared with the graphite particles. Although, if true, the carbon-black binder should be also well identified, which is not the case. What reason could explain silicon is correctly identified, but not the additives?

We do not think that the smaller size of the silicon is the reason. The silicon particles have a very distinct grey value and a clear convex shape. The reason that the CBD is not as accurately identified as silicon lies in the limited resolution of the tomography images.

We have added discussion in the main text (Line 264) on this topic and the benefits of the synthetic data.

*“Training with hybrid learning data also allows the carbon black-binder domain clusters at the edges of silicon particles to be more accurately distinguished (compare **Figure 4d** and **Figure 4f**), with the Dice coefficient for the carbon black-binder domain improving from 0.38 (training of the model with real data only) to 0.58 (training with hybrid data) and the Dice score for silicon improving from 0.69 (only real training data) to 0.82 (hybrid learning data). This improvement can be attributed to the inclusion of different illuminations and edge enhancements in the artificial datasets that are reflective of the variations present at the beamline but that are not fully captured in the limited amount of real training data.”*

- Line 173

[..] such that can the most likely segmentation failure modes are extensively trained [...]

What does mean ‘segmentation failure modes’?

These are the most obvious mistakes in the segmentation. For example, centers of graphite particles are misinterpreted as pore space when the network is trained on real data. To avoid this potential source of confusion for a reader, we add examples into the main text (Line 214).

“For this reason, we turn to computationally generated (i.e., “synthetic” or “artificial”) learning data, which offers control over the data content (i.e., particle sizes and volume fractions) such that the most common segmentation failures (e.g., misinterpreting pore space for graphite) are extensively trained and the neural network is more robust. “

- Line 196

The resulting simulated tomography images (Figure 4f) together with the labeled digital twins form the artificial learning datasets

To better judge of the accurateness of the segmentation problem for these generated structures and associated tomography volumes, compared with the actual ones, I suggest adding a simplified version of figure 2, but applied to these generated tomography scans. Is the histogram still as convoluted? Is the threshold-based method still inaccurate? Significant differences would imply that the training is performed on an intrinsic different set of data. Maybe, with a closer to reality generated tomography, the required size of training data set could be reduced, and/or for a same training data set size, the predictions would be even better.

We added a section comparing real and artificial tomography to the SI (Section 4). It can be seen that the histogram is similarly convoluted and thresholding does not work on the artificial learning data. The main differences between synthetic and real tomography images (as noise level and blur) are by design to add more variance to the learning data.

“When comparing simulated tomography to real tomography images, we reach a structural similarity index (SSIM) (Horé and Ziou 2010) of approximately 0.28. Split up into its components, we find rather high values for the accordance in luminance (≈ 0.92) and contrast (≈ 0.94), and low values (≈ 0.32) for the similarity in structure. The SSIM value however is not used as a target function but to get an overall idea of how to adjust the gray value histogram.

Figure S6: Cross-section of a real tomography image (a), the segmentation applying a k-means algorithm (b), and the corresponding gray value histogram with the four material phases indicated (c). A slice of a simulated tomography image (d), the k-means segmentation (e), and the gray value histogram (f).

Looking at the simulated tomography (Figure S6d) it can be seen that the resolution is lower compared to the real tomography data (Figure S6a). This is due to the fact that the simulated projections are three times smaller in each dimension (width and height of the virtual detector) resulting in a nine times smaller dataset, which is necessary for time-efficient simulation. It is also noticeable that the simulated tomography is less noisy. This is not representative for all simulated images but serves to increase the variation in noise and blur in the training dataset. Furthermore, the gray values of silicon particles and CBD are closer in the simulated case than in the real tomography.

As for the real case, thresholding with a k-means algorithm does not lead to satisfying results for the simulated tomography and the problems are similar to the ones found for the real tomography: mistaking pore space for graphite, assigning particle boundaries to the silicon phase and local illumination differences that make thresholding extremely difficult.

Comparing the gray value histograms of real and simulated tomography (Figure S6c and f) shows an overall correspondence with the main peaks lining up. It can be noticed however that the shoulder assigned to silicon is higher and narrower in the simulated case compared to the real tomography images. Also, there are less saturated pixels. As saturated pixels are mostly located at the top and bottom of the sample and likely to be cut off, this should not present an issue.

Overall, the simulated tomography images match the reality and, most importantly, represent the same difficulties regarding segmentation. Naturally, there is room for improvement as the applied simulation can be seen as a first order approximation to the much more complex wave propagation approach that ideally should be used to simulate phase contrast effects that are of importance for light materials as graphite and carbon black. It has also to be noted that aside from the physics of the tomography the influence of the detector (e.g. saturation limits, normalization) has to be taken into account to ensure comparability.”

- Line 201

The procedure is kept the same as before, except that this time, of 1000 volume pairs, 500 are sampled from real learning datasets and 500 from artificial learning datasets, which is why we refer to it as hybrid learning data.

Training on only real dataset provides inaccurate predictions. What about training on only generated data set? Does it differ from the case where both real and generated data set were used for the training?

We thank the reviewer for this suggestion and added a section on this to the SI (Section 7, Influence of Training Data). As the artificial data is designed to complement the real data and add variation, it is not suited to train a neural network on its own.

“Because training the neural network on the labelled real data, which is available only to a limited amount, leads to unsatisfactory results (Figure S8b) complementary synthetic data is generated. It is specifically designed to add more variation in illumination, noise, and blur as these properties can vary amongst samples of different diameters (the thicker the sample, the more absorbing it is, the higher signal to noise it has) and samples imaged at different times (as the flux at the beamline is not constant over time). However, the synthetic data is not designed to train the network on its own as its variation is too extreme without real data used to “calibrate” the network model. As a consequence, a neural network trained on synthetic data exclusively performs poorly (Figure S8c). Only combining synthetic data with real data, thus resulting in a hybrid training data set, leads to a significantly improved performance (see Figure S8d).

Figure S8: Tomography cross-section (a) with corresponding segmentation stemming from a neural network trained on real learning data (b), synthetic learning data (c), or hybrid learning data (d).

Potentially, one could also generate synthetic data that is very close to the real data and then aim at a neural network that is trained without any real data at all. However, generating synthetic data that looks realistically also requires real data for comparison, which would bring up the question why not to use this real data also to train the neural network.”

- Line 204

Comparing the results from the network trained on hybrid learning data (consisting of 50% of artificial datasets) to the network trained only on the real learning data (as discussed above), the benefits of adding artificial datasets to the learning data can be seen: the quality of the segmentation is shown visually in Figure 5 and assessed numerically with Dice coefficients [46] (Table 1).

- Figure 5d,f,h. The CBD phase seems not very well labelled. 5d seems to neglect the nano porosity, while 5f and 5h under represents the CBD phase – from visual comparison with 5b.

Please see the new discussion in the main text (Line 264) where we describe these points (lack of nanoporosity in 5d and the improvements in 5f and 5h). Note that Figure 4 and 5 have been swapped in the revision.

*“Training with hybrid learning data also allows the carbon black-binder domain clusters at the edges of silicon particles to be more accurately distinguished (compare **Figure 4d** and **Figure 4f**), with the Dice coefficient for the carbon black-binder domain improving from 0.38 (training of the model with real data only) to 0.58 (training with hybrid data) and the Dice score for silicon improving from 0.69 (only real training data) to 0.82 (hybrid learning data). This improvement can be attributed to the inclusion of different illuminations and edge enhancements in the artificial datasets that are reflective of the variations present at the beamline but that are not fully captured in the limited amount of real training data.*

*To further improve the accuracy of the carbon black-binder domain and pore space segmentation, we can add basic thresholding on top of the neural network prediction. For each of the twelve tomograms, we determine an intensity threshold from the gray value histogram corresponding to the boundary between what is identified as graphite and silicon particles, as determined by a k-means approach requiring no manual interaction (see for example **Figure 2d**). We then apply this threshold to all voxels identified as carbon black-binder domain and pore space by the neural networks. Since the carbon black-binder domain contains copper nanoparticles as a staining element which has higher x-ray absorption than silicon, we assign voxels with a gray value above the threshold to the carbon black-binder domain and voxels below the threshold to the pore space. This procedure improves the segmentation of the smaller structural features of the carbon black-binder domain (**Figure 4h**) and leads to a Dice coefficient of 0.72.”*

Expected volume fraction from weight fractions are known (line 280) and could be used to try discriminate the segmentation, although it is not sure the field of view is representative of the volume fractions.

Considering the imaging resolution size may be still too coarse to resolve the CBD nanopore, would it be more reasonable to instead try to catch the micrometer scale representation (as seen in fig d)? The resulting medium could be considered as a heterogeneous medium at this scale, and thus assigned with effective properties (CBD nanoporosity has been quantified at <https://doi.org/10.1021/acsaem.8b00501>).

The citation is added with a descriptive sentence as follows (Line 283):

“While the shape of carbon black-binder domain clusters is now segmented, FIB-SEM imaging indicates that the carbon black-binder domain has an internal porosity of $27\pm 3\%$ [55]. However, this nanoscale porosity internal to the carbon black-binder domain is below the image resolution (119 nm) and therefore is not visible or segmentable.”

- Another issue is the lack of contact between the graphite, silicon, and CBD phases. Phases seem not to touch each other, which prevents estimation of active surface area estimation. Did you calculate interfacial area to verify if contact issue is real or is only not well represented with a 2d slice view. Adding the interfacial area – even with a basic method such as counting the face between voxels belonging to different phases – would provide a relative estimation between the three cases.

We calculated the interface areas (in Figure 8) and they correspond to the ones found in a previous study (Müller et al. 2020).

- Providing a short definition of Dice coefficients could be instructive for non machine learning experts, and how to interpret them (e.g., a dimensional parameter, ranging from 0 to 1, the higher the better, a value of 1 means ***).

We like this suggestion and have added the following (Line 194) :

“We evaluate segmentation quality according to nineteen standard metrics based on similarity and distance criteria [39], where the learning-based segmentations of one pristine sample are compared to the corresponding “ground truth” (Table S2). We focus on the Dice coefficient [39] (Table 1), which is the normalized volumetric overlap of voxels in the segmentation and the “ground truth” of a given phase. A Dice score of zero means no overlap between segmentation and the “ground truth”; a Dice value of one corresponds to complete overlap between segmentation and the “ground truth” segmentation.”

- Line 209

To evaluate and compare the different neural network models, 400 sequential slices of the TXTM dataset of one of the three pristine samples are manually segmented with the help of the Dragonfly software.

Did you compare with segmentations realized with the WEKA fiji plugin? Being free and open source compared with Dragonfly, WEKA is often the first try for segmentation of microstructure too complex for basic thresholding.

We agree that WEKA is also a valuable alternative. The Dragonfly version that was used is also free for academic purposes.

- Line 250

This procedure improves the segmentation of the internal structure carbon black-binder domain (Figure 5h) [...]

This is debatable, see previous comments.

Looking at the Dice coefficient and further metrics (SI Table S2), the subsequent thresholding improves the accuracy of the CBD segmentation.

Table S2: Segmentation evaluation measures.

Metrics	Pore Space			Graphite			Silicon			CBD			Average			Weighted Average		
	Real learn. data	Hybrid learn. data	Hybrid learn. data + thresh.	Real learn. data	Hybrid learn. data	Hybrid learn. data + thresh.	Real learn. data	Hybrid learn. data	Hybrid learn. data + thresh.	Real learn. data	Hybrid learn. data	Hybrid learn. data + thresh.	Real learn. data	Hybrid learn. data	Hybrid learn. data + thresh.	Real learn. data	Hybrid learn. data	Hybrid learn. data + thresh.
Similarity																		
Dice Coefficient	0.626	0.692	0.715	0.645	0.769	0.769	0.688	0.816	0.816	0.381	0.582	0.719	0.585	0.715	0.755	0.626	0.727	0.744
Jaccard Coefficient	0.456	0.529	0.557	0.476	0.624	0.624	0.525	0.689	0.689	0.235	0.411	0.561	0.423	0.563	0.608	0.457	0.573	0.593
Area under ROC Curve	0.654	0.735	0.754	0.702	0.773	0.773	0.772	0.892	0.892	0.865	0.842	0.906	0.749	0.811	0.831	0.690	0.763	0.775
Cohen Kappa	0.307	0.478	0.515	0.411	0.542	0.542	0.680	0.810	0.810	0.333	0.558	0.703	0.433	0.597	0.642	0.369	0.523	0.546
Rand Index	0.548	0.620	0.638	0.589	0.646	0.646	0.967	0.976	0.976	0.773	0.908	0.941	0.719	0.788	0.800	0.592	0.658	0.667
Adjusted Rand Index	0.097	0.240	0.276	0.178	0.292	0.292	0.668	0.799	0.799	0.279	0.528	0.679	0.305	0.465	0.512	0.163	0.297	0.320
Interclass Correlation	0.625	0.691	0.714	0.644	0.768	0.768	0.688	0.816	0.816	0.381	0.582	0.719	0.584	0.714	0.754	0.625	0.726	0.743
Volumetric Sim. Coeff.	0.982	0.905	0.914	0.852	0.941	0.941	0.740	0.965	0.965	0.443	0.807	0.860	0.754	0.905	0.920	0.888	0.919	0.926
Mutual Information	0.069	0.176	0.204	0.134	0.228	0.228	0.092	0.133	0.133	0.085	0.104	0.149	0.095	0.160	0.179	0.101	0.195	0.210
Distance																		
Hausdorff Distance**	82.80*	52.40*	52.40*	72.30*	75.60*	73.20*	258.6	156.1	156.1	56.80*	70.10	57.70	117.6*	88.60*	84.90*	82.60*	67.60*	65.90*
Average Hausdorff Dist.**	5.187*	1.349*	1.302*	3.288*	2.197*	1.531*	8.468	1.575	1.575	1.661*	1.331	0.806	4.700*	1.613*	1.303*	4.245*	1.725*	1.395*
Mahanobolis Dist.**	0.400	0.146	0.101	0.321	0.085	0.085	0.216	0.062	0.062	0.273*	0.358	0.080	0.234*	0.163*	0.082*	0.351*	0.125*	0.091*
Variation of Information	1.852	1.596	1.544	1.661	1.539	1.539	0.170	0.150	0.150	0.745	0.425	0.308	1.107	0.927	0.885	1.654	1.465	1.437
Global Consistency Error	0.569	0.434	0.409	0.475	0.405	0.405	0.027	0.023	0.023	0.167	0.078	0.053	0.309	0.235	0.222	0.488	0.390	0.377
Probabilistic Distance	0.002	0.002	0.002	0.002	0.001	0.001	0.002	0.001	0.001	0.006	0.003	0.002	0.003	0.002	0.001	0.002	0.002	0.001
Classic Measures																		
Sensitivity	0.638	0.631	0.659	0.562	0.817	0.817	0.546	0.789	0.789	0.861	0.722	0.836	0.652	0.740	0.775	0.610	0.728	0.745
Specificity	0.670	0.839	0.849	0.842	0.729	0.729	0.999	0.995	0.995	0.870	0.963	0.976	0.845	0.882	0.887	0.771	0.799	0.804
Precision (Confidence)	0.615	0.764	0.783	0.757	0.726	0.726	0.930	0.845	0.845	0.245	0.488	0.630	0.637	0.706	0.746	0.675	0.736	0.751
Accuracy	0.656	0.745	0.763	0.711	0.770	0.770	0.983	0.988	0.988	0.870	0.952	0.969	0.805	0.864	0.873	0.703	0.775	0.784

*due to memory limitations, the volume is split into four quarters (250x200x576 voxel each) and the metric is calculated independently on each quarter and then averaged.

**the value is reported in voxels, with 1 voxel=61 nm

- Line 262

The cylindrical tomography images are binned by a factor of 2.22 resulting in a voxel size of 61 nm.

Additives nanopore size range from 5 to 150 nm [*]. You could include this information as it provide some number to explain why the CBD small features are difficulty segmentable.

[*] L. Zielke, T. Hutzenlau, D. R. Wheeler, C.-Wei Chao, I. Manke, A. Hilger, N. Paust, R. Zengerle, and S. Thiele, Three-Phase Multiscale Modeling of a LiCoO₂ Cathode: Combining the Advantages of FIB–SEM Imaging and X-Ray Tomography, *Adv. Energy Mater.* 2015, 5, 1401612

We thank the reviewer for the suggestion and refer to line 59 where the citation has already been incorporated.

- 282 Microstructural Analysis and Changes with Cycling

The evolution analysis is limited by the small field of view (representativity issue), thus establishing trends is risky, as, as indicated in the text “we are not tracking the microstructural evolution of the same sub-section of electrode with cycling”. There is a real possibility of over-interpreting microstructure parameter evolution with cycling in this section. Then “we therefore focus on robust trends across the three samples and detailed analysis of changes around specific phases” seems a too strong statement, especially considering only 3 small samples are considered.

This sentence was not claiming that we can analyze everything but rather a statement that we focus only on parameters that can be analyzed.

To make this clearer, we changed the statement to read as follows (Line 319):

“...given the small sample size and the resulting compositional variations across the three samples (Figure 7a), we therefore focus on trends that hold for all three samples and detailed analysis of changes around phases (e.g. the changes in the carbon black-binder domain around graphite or silicon active particles).”

- Line 296

The exact number may be a slight underestimate since the identification of graphite particles during segmentation is based on edge-enhancement of the tomography data (see Figure 2a), making it difficult to distinguish bright graphite edges and the bright carbon-black and binder domain signal.

“may be a strong underestimate” seems more realistic.

We agree with the reviewer, that “slight” might give a wrong impression here, however, as the phase fraction analysis suggests, that the volume fractions are quite accurate and therefore we do not think that we “strongly” underestimate the surface coverage. We changed the statement to (Line 331):

“This value may be an underestimate since the identification of graphite particles during segmentation is based on edge-enhancement of the tomography data (see Figure 2a), making it difficult to distinguish bright graphite edges and the bright carbon-black and binder domain signal.”

- Line 313

We perform numerical diffusion simulations on the electrode structures and find that the effective transport coefficient is 0.14.

Please explain how you realized the calculation briefly (Tau factor?) and the choice of the boundary conditions. Volume being small, the calculation may suffer from edge effects (isolated pores located at the domain edges, named “unknown cluster”, bias the calculation for low volumes). As well, for low volumes, the choice of the boundary conditions will impact the solution. A possible way to increase the level of confidence in the result would be calculate the effective transport coefficient with Dirichlet-Dirichlet boundary conditions and then again but with Neumann-Neumann (with an additional fixed point) to verify the two bounds are not far from each other. See J. Joos et al., J. Power Sources, 246, 819 (2014) for the concept of unknown clusters, and F. Usseglio-Viretta et al., ECS Transactions, 77 (11) 1095-1118 (2017) for the impact of boundary conditions.

We thank the reviewer for his comment. The simulation is explained in the Methods section in the manuscript. The effective transport coefficient is calculated from the simulated effective diffusivity ($\delta = \frac{D_{eff}}{D}$). It would also be possible to calculate it from the tortuosity factor; however, this will lead to similar results, as the tortuosity factor is also calculated via the effective diffusivity. We used symmetric boundary conditions as we have a comparably low porosity and a small sample size.

Given the uncertainty of the CBD segmentation (cf. Figure 5), the diffusion coefficient should be calculated with the CBD phase obtained with figures 5d and 5h to provide uncertainty bounds. For both cases, the graphite segmented with the hybrid learning data + thresholding would be used. The same approach could also be used for the interfacial area calculations.

Calculating the diffusion coefficient with less accurate segmentation will not tell us the uncertainty in the segmentation. We now include 19 standard metrics for quantifying the quality of the segmentation (Table S2) and note the main reason for uncertainty (the images have a resolution of 119 nm and therefore porosity internal to the CBD cannot be resolved).

Of interest also would be to calculate the diffusion coefficient when CBD and/or silicon are not represented to have an estimation of the error performed when only graphite and pore are considered, as they are the phases the most simple to segment. This is what you did (line 320), although what is the evolution of the exponential coefficient alpha (tortuosity=porosity(1-alpha)) between the two cases? Alpha is likely to increase when the CBD is added to the analysis, indicating that the CBD impact on tortuosity is more than just a reduction of porosity. Such result is important as it indicates one cannot simply extrapolate a tortuosity-porosity relationship established on a graphite pore domain to a graphite CBD pore domain, as previously investigated in [23].

The authors would like to note that graphite and pore space are actually the hardest phases to segment due to the low contrast in gray value. We added the Bruggeman exponent (SI Section 9):

“Macroscopically, the tortuosity τ is related to the porosity ϵ of porous electrodes by the Bruggeman exponent α and has been studied for porous electrode microstructures (Ebner et al. 2014).

$$\tau = \epsilon^{-\alpha}$$

The fact that the Bruggeman exponent calculated on the electrode structures with the CBD included is smaller than the exponent calculated on the structures neglecting the CBD (Table S3) indicates that the increase in the tortuosity brought about by CBD does not only come from a decrease in porosity but that the CBD also plots pores, creating more tortuous transport.”

Table S3: Bruggeman exponent calculated for the different microstructure including and neglecting the CBD.

Bruggeman exponent α [-]	Number of cycles				
		pristine	2	5	8
Including CBD		1.78 ± 0.24	1.67 ± 0.22	1.67 ± 0.11	1.77 ± 0.11
Neglecting CBD		1.93 ± 0.36	1.77 ± 0.31	1.75 ± 0.15	1.92 ± 0.16

- Line 330

Nonetheless, quantification of the morphology of the carbon black-binder domain is one step towards understanding ionic and electronic performance of the electrode and it can be used to form the basis of computer-generated carbon black-binder domains [34][54][55][56][*]

[*] A. N. Mistry, K. Smith, and P. P. Mukherjee, Secondary-Phase Stochastics in Lithium-Ion Battery Electrodes, ACS Appl. Mater. Interfaces 2018, 10, 7, 6317–6326

We thank the reviewer for pointing out this publication and we added the citation (Reference [58]).

- Line 365

This means that trends induced by the cycling behavior of the silicon can easily be hidden within the statistical variation found sample to sample, and would require a larger field of view to reach the representativity scale of the silicon spatial distribution.

As mentioned already in a previous comment, the low volume fraction of silicon in the samples make the representative volume for silicon larger. We agree with the reviewer and added the suggested subsentence (Line 389).

“Trends induced by the cycling behavior of the silicon can easily be hidden within the statistical variation found sample to sample, and would require a more and/or larger volumes to be analyzed in order to account for the variability.”

- Line 381

The increase in the amount of carbon black-binder domain in subsequent volume shells (Figure 8d) can be explained by the carbon black-binder domain forming clusters in the pore space between active particles.

Very interesting result that should be stressed more as it can guide development of CBD generating algorithms. Some CBD generation algorithm such as the one from Hein [56] and Usseglio [*] are using a ‘bridge’ approach, assuming CBD is likely to be between neighbored particles, which could fit with this result, as CBD must be present in the pore space between particles. Other algorithm, that tends to generate CBD preferentially at the particle surface (thus creating a thin layer at the surface with more or less surface rugosity) would less fit this result.

[* no need to be quoted, but FYI] MATBOX: Microstructure Analysis Toolbox, <https://github.com/NREL/MATBOX> Microstructure analysis toolbox

- Line 386

Here the opposite trend is found, indicating CBD generation algorithms that favor deposition at the particle surface are more suited for silicon, compared with CBD generation algorithms built around a 'bridge' approach.

Results suggests combining different CBD generation algorithms/methods may be required to adequality model CBD spatial distribution in graphite silicon electrodes.

The reviewer makes an excellent point with these two comments. We have now added this to the discussion in the main text (Line 421).

"The statistical analysis of structure enabled by our segmented reconstructions leads to several findings. First, distribution of the carbon black-binder around graphite particles and around silicon particles differs, perhaps due to differences in the adhesion of the carbon black-binder to the two particle types or the local differences in the slurry mixing (e.g., shear forces). Such different distributions certainly impact the mechanical and electrochemical properties of the electrode, and this serves as an important reminder when computationally generating carbon black-binder structures for simulation purposes that, in composite electrodes, the carbon black-binder domain may look structurally different in sub-regions."

- Supplemental information

7. Additional Structural Analysis

How are diameters calculated? There is a whole family of numerical methods to calculate a diameter.

Knowing the method help better judge the results.

We agree that this information is important, as there are many different methods to calculate the pore diameter and the CBD thickness respectively. We added this information to the SI.

"We used the PoroDict toolbox of the GeoDict2020 Software (Math2Market GmbH, Kaiserslauten, Germany) and calculated the pore size distribution and the CBD thickness by granulometry with symmetric boundary conditions."

Reviewer 2

This manuscript pertains to the application of deep learning techniques, namely an adaptation of the 3D-UNet convolutional neural network for segmentation of four-phase battery electrodes consisting of graphite, silicon, CBD, and void. Yet beyond the application of DL for segmentation, the paper is incredibly broad, representing the manufacturing, cycling, and imaging of electrodes, plus the study of a variety of segmentation techniques (both manual and twists on the traditional DL processes). This manuscript represents the compilation of a significant amount of work, but with the focus on the very important topic of image segmentation. In particular, this work represents the first segmentation of a battery electrode that includes image-based representation of the CBD at a scale and resolution that is usable for 3D mesoscale simulation. While the methods that are highlighted in the manuscript are worthy on their own, the image datasets that are promised for publication will be widely used by many other researchers. As such, I feel that it warrants publication in Nature Communications, after a few minor modifications are made and questions answered, which are listed below.

We thank the reviewer for this positive review.

1. Line 73. You state that DL has not been applied to battery electrodes. Yet there have been two recent publications in this area, one in this journal (doi: [10.1038/s41467-020-16233-5](https://doi.org/10.1038/s41467-020-16233-5)) and another preprint from last year (arxiv: 1910.10793), both of which apply CNNs to cathode and anode images, respectively.

We cite these two papers [30],[32],and rephrase our sentence (Line 67) to read:

*“While deep learning algorithms have been applied to assess the state of health of a battery [24], to improve the microstructural design [25], [26], or to detect defects during manufacturing [27], microstructure segmentation continues to rely heavily on adapted filtering followed by simple thresholding operations [28]. Even though algorithms originating in the field of machine learning are beginning to be used for segmentation [29], [30], crack detection [31] and particle detachment [32] to date, deep learning-based approaches are mainly used on high-contrast systems (i.e. cathodes), and algorithms for complete 3D segmentation such as those which are used in medicine to analyze tomographic full-body scans and identify organs [33], have not been applied to semantic segmentation of LIB electrode datasets.”We note that citation * is not reporting a complete segmentation, but rather using machine learning is used to segment NMC particles (to identify cracked particles as one particle).*

2. To obtain segmentations for a large domain, you create smaller subdomains and “average” their results in the overlapped region using a radial weighting function. Do you also do this during training, or just at inference time? Using this overlap during training could be particularly important because of the small amount of data that can fit in CPU memory; you specifically call out that some graphite particles can span the training subdomains (line 235).

Overlap is also used during training. We now explicitly state this in the main text when describing the learning data (Line 177):

“Due to memory limitations, 1000 subvolume pairs (the 3D image data and label map) of dimensions 15.6x15.6x3 μm (256x256x48 voxel) are randomly (and thus potentially overlapping) sampled from the learning data such that the smallest dimension is equally likely to be along the x, y, or z axis. 50% of the training data is subjected to data augmentation based on basic image manipulations in order to

render the neural network more robust and increase its ability to generalize while decreasing the potential danger of overfitting [51]. Specifically, we apply Gaussian noise and blurring, vary contrast and brightness, and use affine transformations. The model is trained for 100 epochs, which amounts to a total training time of 72 hours on a Nvidia TITAN Xp GPU.”

Indeed, we state in the main text (Line 256) the benefits of the training data and its limitations due to the limited size:

“...artificial datasets are designed to be representative of the volume percents of each phase and the particle size distributions and contain more large graphite particles than the real datasets, which are often randomly cut in the sample preparation prior to imaging. Nonetheless, segmentation of the graphite and pore phases remains a challenge because of the low contrast between the graphite and pore space, the lack of distinguishing features within bulk graphite, and imaging artefacts around the edges of small graphite particles. Furthermore, the size of the receptive field [55] (i.e. the subset of the input image that influences the prediction for a certain voxel), is at times smaller than a graphite particle.”

3. Your “basic structure” generation algorithm seems rather ad-hoc, especially given how researchers have recently used physics-based simulations to generate microstructures (see doi: 10.1016/j.jpowsour.2019.227285 and doi: 10.1021/acsami.0c08251 for examples). Now, it may not matter that much because of your application of CycleGAN afterwards. But you should better justify your present approach and at least acknowledge the increased fidelity of simulated mesostructures in the literature.

These papers are now cited [53],[54] and the possibility of simulated mesostructures acknowledged (Line 225).

4. Regarding the CycleGAN, there was a nice paper recently published in this journal family (doi: 10.1038/s41524-020-0340-7) that pioneered using CycleGANs in the way that you describe, that should be cited here (around line 184).

This is now cited as Reference [45].

5. Line 199: Why are your artificial learning datasets so much smaller in size than the image-based datasets? Do you think this influences the quality of the model?

The artificial datasets are limited in volume due to the memory limitations of CycleGAN and 3D U-Net. We now extensively discuss the improvements the artificial learning data still brings over the real data (e.g., representative particle sizes and volume presents as well as the variations in illumination and edge effects) and we quantify the impact this has on the accuracy of the simulation.

6. Given the difficulty in creating manual segmentations for the domains, how good do you think your Dice score comparisons are? Could you instead compare to the “hybrid” data, both the manually segmented and generated images?

While manual segmentation is quite tedious and therefore not practical to implement for a large number of datasets, the human eye is very reliable in spotting features. Therefore, manual segmentation is a reasonable ground truth.

7. It appears that all of your quantitative metrics for image segmentation quality are on the training domain and none on held-out samples. It is typical in the ML community to not only compare model accuracy in the training domain, but also in held-out samples that have also been manually segmented. It is common for models to perform very well on the training domain, but less-so on new, held-out data.

No, we do check modal accuracy based on a held-out sample. To make this clear in the maintext, we write (Line 193):

“This manually segmented volume serves as the “ground truth” and is otherwise not used for training. We evaluate segmentation quality according to nineteen standard metrics based on similarity and distance criteria [40], where the learning-based segmentations of one pristine sample are compared to the corresponding “ground truth” (Table S2).”

8. Line 333: Consider reference the work of Franco and coworkers, specifically doi: 10.1016/j.jpowsour.2019.227285.

This is now cited as Reference [53].

9. Line 353: You say that the CBD does not deform extensively. Do you mean it does not plastically/inelastically deform? I suspect that during a charge-discharge cycle, it does deform more significantly.

Yes, we mean that the structure is not deformed. Since this is not an operando experiment, we cannot visualize how the CBD may be changing during a cycle. However, the segmentation could allow us to simulate what is going on.

We now state this more clearly (Line 337): “these values do not change after cycling, highlighting that cycling does not induce a permanent change in the morphology of the carbon black-binder domain in the vicinity of a graphite particle.”

10. You use the phrase “ground truth” to denote the manually segmented image data used for training. Typically, however, “ground truth” refers to training (or validation) data that is known to be exactly correct. Your manual segmentations, while likely good for training, are not perfectly true. You should use “training data” rather than “ground truth”.

We do not use a manually segmented image for training so certainly it should not be called training data. We use the manually segmented image in order to quantify the accuracy of our segmentation. We use “ground truth” in quotations exactly because there is no way of having an exactly correct segmentation.

11. SI Section 3.1: The porosity of 5-40% of CBD seems rather low compared to doi: 10.1002/aenm.201401612.

This is only used for the “basic structure”, CycleGAN takes care of the remaining difference. We now state this explicitly in the SI.

12. SI Section 6: You refer to the output of the network as a “probability map.” You do not have

enough details of the network here to verify, but typically these networks output a softmax layer, which is not to be exactly interpreted as a probability map.

We had used the term probability map to make it more understandable for a reader from the battery community, but you are correct that we should be precise. We now use the term “softmax”, also in the main text (Line 305):

*“For each subvolume, a softmax representation is created, which corresponds to a map, where each voxel contains the probability of it belonging to each material phase (pore space, graphite, silicon, or carbon black-binder domain). As described in the **Supplementary Information Note 6**, the probability maps (softmax representations) from each sampling are then combined to obtain the final segmentations (**Figure 6b**).”*

Reviewer #3 (Remarks to the Author):

This work used deep Learning-based approach to analyze the X-ray tomographic data, and applied the technique for Li-ion batteries. As the authors stated, high-quality images are either technically not possible or not affordable from a time or resource perspective. In such situations, deep learning-based method can be very helpful for accurate segmentation of volumetric datasets. The authors showed the morphological evolution of graphite-silicon anodes, particularly carbon black-binder domain detached at regions close to the silicon particles. These are helpful to understand the impact on transport.

While the key part of this work is about machine learning, the machine learning approach itself was not new (borrowed a specific implementation of the 3D-Unet architecture from Baumgartner et al.). The approach has been well applied for medical imaging already. Since this work is machine learning focused while the work itself did not provide sufficient advances on machine learning, it would fit better a more applied journal.

The goal of our work is not to develop new machine learning algorithms, but that rather to show a way that existing tools can be applied in a systematic approach to volumetric images of battery electrodes. To make this clearer, we have revised our abstract to explicit mention that are using the 3D U-Net architecture and highlight the novelty:

“Accurate 3D representations of lithium ion battery electrodes, in which the active particles, binder and pore phases are distinguished and labeled, can assist in understanding and ultimately improving battery performance. Here, we demonstrate a methodology for using deep-learning tools to achieve reliable segmentations of volumetric images of electrodes on which standard segmentation approaches fail due to insufficient contrast. We implement the 3D U-Net architecture for segmentation, and, to overcome the limitations of training data obtained experimentally through imaging, we show how synthetic learning data, consisting of digital twins of our electrodes and their tomographic reconstructions, can be generated and used to enhance network performance. We apply our method to segment x-ray tomographic microscopy images of silicon-graphite composite electrodes and show it is accurate across standard metrics. We then apply it to obtain a statistically meaningful analysis of the microstructural evolution of the carbon-black and binder domain during battery operation.”

In addition, the authors have previous published papers such as “Multimodal nanoscale tomographic imaging for battery electrodes”, which shows a more comprehensive study of graphite-silicon

composite electrodes based on tomographic imaging. Thus, results on tomographic imaging of silicon particles do not represent major advances.

Yes, we have published previous work on tomography of graphite-silicon composite electrodes. The previous work focused on multimodal imaging, specifically how to combine scanning electron microscopy and x-ray transmission microscopy or x-ray ptychography and transmission microscopy that offer different resolutions and gray scale contrast. Combining multiple imaging modalities allows high quality image segmentation but is not practical from a time and resource perspective. This is one the motivation for our current work, but our current work is not focused on multimodal imaging. It is about how deep learning approaches can be applied by segmentation of battery electrode and how this can be enhanced by the creation of digital twins.

On the technical side, the following is suggested.

1. No mathematical analysis was provided on the machine learning approach. Such analysis should be discussed.

We explain that we choose an existing architecture from Baumgartner et al.. The authors already provided a mathematic analysis. Our implementation is based on their 3D U-Net architecture, which is the current state-of-the-art. We write (Line 161):

“... We therefore turn to deep learning algorithms, which have entered the field of volumetric image segmentation through the implementation of 3D convolutional networks [48]. Prominent architectures that apply these principles for volumetric segmentation are the V-Net [49] and, more widely used, the 3D U-Net [33]. We borrow a specific implementation of the 3D U-Net architecture from Baumgartner et al., where the authors compared a variety of architectural choices for semantic segmentation of volumetric medical images acquired with magnetic resonance imaging (MRI) [50].”

2. Rigorous quantification of the machine learning performance should be provided.

We have now added Table S2, which provides quantification of the machine learning based segmentation according to 19 standard metrics.

3. A comparison to other benchmark techniques should be provided.

Figure 2 shows the limitations of k-means and random walk algorithms, which are the other 2 standard approaches used in the 3D battery image segmentation approach.

REVIEWER COMMENTS

Reviewer #1 (Remarks to the Author):

I thank the authors to have addressed most of the comments issued with the first submission. The article gained in context with addition of ML segmentation review and readability with numerous details/explanations added to further explain methods and results. The added summary on the findings of the CBD analysis (line 420) is very valuable to guide generation algorithm too.

In overall, the work is very significant and helpful and provide valuable insights on the silicon/graphite/CBD morphology and spatial distribution, in addition to demonstrate an efficient segmentation methodology. My only concerns lie on (i) the probable lack of representativity due to the limited field of view, that prevent quantification for macroscale application/modeling - mentioned in the text, e.g. line 393) and (ii) the difficulty to assess interfacial area - mentioned in the text too, line 334.

One minor comment:

Line 354

Previous comment on the effective diffusion coefficient has not been fully addressed. The 0.14 value for the effective diffusion coefficient has potentially a high uncertainty due to limited field of view. Calculation with different boundary conditions would help to bound the value. Suggestion below: We perform numerical diffusion simulations on the electrode structures and find that the effective transport coefficient is 0.14. However, given the relatively small field of view, this value should be considered with caution, as uncertainty due to lack of representativity and edge effects are expected to be high.

One typo error:

Line 393

“in order to”

Reviewer #2 (Remarks to the Author):

This manuscript pertains to the application of deep learning techniques, namely an adaptation of the 3D-UNet convolutional neural network for segmentation of four-phase battery electrodes consisting of graphite, silicon, CBD, and void. Yet beyond the application of DL for segmentation, the paper is incredibly broad, representing the manufacturing, cycling, and imaging of electrodes, plus the study of a variety of segmentation techniques (both manual and twists on the traditional DL processes). This manuscript represents the compilation of a significant amount of work, but with the focus on the very important topic of image segmentation. In particular, this work represents the first segmentation of a battery electrode that includes image-based representation of the CBD at a scale and resolution that is usable for 3D mesoscale simulation. While the methods that are highlighted in the manuscript are worthy on their own, the image datasets that are promised for publication will be widely used by many other researchers. As such, I feel that it warrants publication in Nature Communications, after a few minor modifications are made and questions answered, which are listed below.

1. Line 73. You state that DL has not been applied to battery electrodes. Yet there have been two recent publications this area, one in this journal (doi: 10.1038/s41467-020-16233-5) and another preprint from last year (arxiv: 1910.10793), both of which apply CNNs to cathode and anode images,

respectively.

2. To obtain segmentations for a large domain, you create smaller subdomains and “average” their results in the overlapped region using a radial weighting function. Do you also do this during training, or just at inference time? Using this overlap during training could be particularly important because of the small amount of data that can fit in CPU memory; you specifically call out that some graphite particles can span the training subdomains (line 235).

3. Your “basic structure” generation algorithm seems rather ad-hoc, especially given how researchers have recently used physics-based simulations to generate microstructures (see doi: 10.1016/j.jpowsour.2019.227285 and doi: 10.1021/acsami.0c08251 for examples). Now, it may not matter that much because of your application of CycleGAN afterwards. But you should better justify your present approach and at least acknowledge the increased fidelity of simulated mesostructures in the literature.

4. Regarding the CycleGAN, there was a nice paper recently published in this journal family (doi: 10.1038/s41524-020-0340-7) that pioneered using CycleGANs in the way that you describe, that should be cited here (around line 184).

5. Line 199: Why are your artificial learning datasets so much smaller in size than the image-based datasets? Do you think this influences the quality of the model?

6. Given the difficulty in creating manual segmentations for the domains, how good do you think your Dice score comparisons are? Could you instead compare to the “hybrid” data, both the manually segmented and generated images?

7. It appears that all of your quantitative metrics for image segmentation quality are on the training domain and none on held-out samples. It is typical in the ML community to not only compare model accuracy in the training domain, but also in held-out samples that have also been manually segmented. It is common for models to perform very well on the training domain, but less-so on new, held-out data.

8. Line 333: Consider reference the work of Franco and coworkers, specifically doi: 10.1016/j.jpowsour.2019.227285.

9. Line 353: You say that the CBD does not deform extensively. Do you mean it does not plastically/inelastically deform? I suspect that during a charge-discharge cycle, it does deform more significantly.

10. You use the phrase “ground truth” to denote the manually segmented image data used for training. Typically, however, “ground truth” refers to training (or validation) data that is known to be exactly correct. Your manual segmentations, while likely good for training, are not perfectly true. You should use “training data” rather than “ground truth”.

11. SI Section 3.1: The porosity of 5-40% of CBD seems rather low compared to doi: 10.1002/aenm.201401612.

12. SI Section 6: You refer to the output of the network as a “probability map.” You do not have enough details of the network here to verify, but typically these networks output a softmax layer, which is not to be exactly interpreted as a probability map.

13. Line 354 the authors reference numerical diffusion simulations, but no details on this calculation or the input parameters are given.

14. The methods section is overall very lacking. There are very few details of any of the numerical procedures, which are warranted. Even on the segmentation aspects, more details are required (such as how you did the manual segmentation in Dragonfly).

Reply to the reviewer comments:

We thank the reviewers for their detailed feedback on the paper and their insightful comments. Below, we present the point-by-point reply to the comments, questions and suggestions.

Reviewer #1 (Remarks to the Author):

I thank the authors to have addressed most of the comments issued with the first submission. The article gained in context with addition of ML segmentation review and readability with numerous details/explanations added to further explain methods and results. The added summary on the findings of the CBD analysis (line 420) is very valuable to guide generation algorithm too.

In overall, the work is very significant and helpful and provide valuable insights on the silicon/graphite/CBD morphology and spatial distribution, in addition to demonstrate an efficient segmentation methodology. My only concerns lie on (i) the probable lack of representativity due to the limited field of view, that prevent quantification for macroscale application/modeling - mentioned in the text, e.g. line 393) and (ii) the difficulty to assess interfacial area - mentioned in the text too, line 334.

We thank the reviewer for their positive feedback as well as the helpful comments.

As we mention in the main text, we agree that an increased field of view would be desirable; however, it would result in lower resolution and/or increased imaging time, which was unfeasible for this study.

We have revised the description of the simulations to get the effective transport parameters and implemented this in the main text and the supplementary information (SI). Please find the detailed response to the comment below.

One minor comment:

Line 354

Previous comment on the effective diffusion coefficient has not been fully addressed. The 0.14 value for the effective diffusion coefficient has potentially a high uncertainty due to limited field of view. Calculation with different boundary conditions would help to bound the value. Suggestion below:

We perform numerical diffusion simulations on the electrode structures and find that the effective transport coefficient is 0.14. However, given the relatively small field of view, this value should be considered with caution, as uncertainty due to lack of representativity and edge effects are expected to be high.

We agree with the reviewer that the value of the diffusivity should be considered with caution. We use symmetric boundary conditions in order to decrease sample edge effects. Table 2 also gives the values for the transport coefficient including error.

Our small sample size can lead to a lack of representability; however, the effective transport coefficient of 0.14 is in the range of values from comparable studies (i.e. Pietsch et. al, Sustain. Energy Fuels (2018)) and we are careful in our analysis and only look for trends with cycling in sub-volumes of the electrodes with similar volume fractions of the different phases.

We added a detailed description of the diffusion simulation in the method section. It reads:

“Both border planes (along the through-plane direction) are set to have a constant concentration which decreases edge effects. Symmetric boundary conditions are also applied in all tangential directions and the Neumann (zero flux) condition is used at the domain boundaries. The concentration drop across the through-plane direction was set to 1 and the (dimensionless) Laplace equation is solved in an iterative process by finite volume-based solver (EJ) and stops if the process becomes stationary (tolerance of 0.001).

One typo error:

Line 393

“in order to”

We thank the reviewer for pointing this out and we corrected the typo.

Reviewer #2 (Remarks to the Author):

This manuscript pertains to the application of deep learning techniques, namely an adaptation of the 3D-UNet convolutional neural network for segmentation of four-phase battery electrodes consisting of graphite, silicon, CBD, and void. Yet beyond the application of DL for segmentation, the paper is incredibly broad, representing the manufacturing, cycling, and imaging of electrodes, plus the study of a variety of segmentation techniques (both manual and twists on the traditional DL processes). This manuscript represents the compilation of a significant amount of work, but with the focus on the very important topic of image segmentation. In particular, this work represents the first segmentation of a battery electrode that includes image-based representation of the CBD at a scale and resolution that is usable for 3D mesoscale simulation. While the methods that are highlighted in the manuscript are worthy on their own, the image datasets that are promised for publication will be widely used by many other researchers. As such, I feel that it warrants publication in Nature Communications, after a few minor modifications are made and questions answered, which are listed below.

We thank the reviewer for this positive review. The questions and comments (1-12) below have been addressed and implemented already in the first submission round.

1. Line 73. You state that DL has not been applied to battery electrodes. Yet there have been two recent publications this area, one in this journal (doi: 10.1038/s41467-020-16233-5) and another preprint from last year (arxiv: 1910.10793), both of which apply CNNs to cathode and anode images, respectively.

We cite these two papers [30],[32] and rephrase our sentence (Line 67) to read:

“While deep learning algorithms have been applied to assess the state of health of a battery², to improve the microstructural design^{3,4}, or to detect defects during manufacturing⁵, microstructure segmentation continues to rely heavily on adapted filtering followed by simple thresholding operations⁶. Even though algorithms originating in the field of machine learning are beginning to be used for segmentation^{7,8}, crack detection⁹ and particle detachment¹⁰ to date, deep learning-based approaches are mainly used on high-contrast systems (i.e. cathodes), and algorithms for complete 3D segmentation such as those which are used in medicine to analyze tomographic full-body scans and identify organs¹¹, have not been applied to semantic segmentation of LIB electrode datasets.”

2. To obtain segmentations for a large domain, you create smaller subdomains and “average” their results in the overlapped region using a radial weighting function. Do you also do this during training, or just at inference time? Using this overlap during training could be particularly important because of the small amount of data that can fit in CPU memory; you specifically call out that some graphite particles can span the training subdomains (line 235).

Overlap is also used during training. We now explicitly state this in the main text when describing the learning data (Line 177):

“Due to memory limitations, 1000 subvolume pairs (the 3D image data and label map) of dimensions 15.6x15.6x3 μm (256x256x48 voxel) are randomly (and thus potentially overlapping) sampled from the learning data such that the smallest dimension is equally likely to be along the x, y, or z axis. 50% of the training data is subjected to data augmentation based on basic image manipulations in order to render the neural network more robust and increase its ability to generalize while decreasing the potential danger of overfitting¹². Specifically, we apply Gaussian noise and blurring, vary contrast and brightness, and use affine transformations. The model is trained for 100 epochs, which amounts to a total training time of 72 hours on a Nvidia TITAN Xp GPU.”

Indeed, we state in the main text (Line 256) the benefits of the training data and its limitations due to the limited size:

“...artificial datasets are designed to be representative of the volume percents of each phase and the particle size distributions and contain more large graphite particles than the real datasets, which are often randomly cut in the sample preparation prior to imaging. Nonetheless, segmentation of the graphite and pore phases remains a challenge because of the low contrast between the graphite and pore space, the lack of distinguishing features within bulk graphite, and imaging artefacts around the edges of small graphite particles. Furthermore, the size of the receptive field [55] (i.e. the subset of the input image that influences the prediction for a certain voxel), is at times smaller than a graphite particle.”

3. Your “basic structure” generation algorithm seems rather ad-hoc, especially given how researchers have recently used physics-based simulations to generate microstructures (see doi: 10.1016/j.jpowsour.2019.227285 and doi: 10.1021/acsami.0c08251 for examples). Now, it may not matter that much because of your application of CycleGAN afterwards. But you should better justify your present approach and at least acknowledge the increased fidelity of simulated mesostructures in the literature.

These papers are now cited [53],[54] and the possibility of simulated mesostructures acknowledged (Line 225).

4. Regarding the CycleGAN, there was a nice paper recently published in this journal family (doi: 10.1038/s41524-020-0340-7) that pioneered using CycleGANs in the way that you describe, that should be cited here (around line 184).

This is now cited as Reference [45].

5. Line 199: Why are your artificial learning datasets so much smaller in size than the image-based datasets? Do you think this influences the quality of the model?

The artificial datasets are limited in volume due to the memory limitations of CycleGAN and 3D U-Net. We now extensively discuss the improvements the artificial learning data still

brings over the real data (e.g., representative particle sizes and volume presents as well as the variations in illumination and edge effects) and we quantify the impact this has on the accuracy of the simulation.

6. Given the difficulty in creating manual segmentations for the domains, how good do you think your Dice score comparisons are? Could you instead compare to the “hybrid” data, both the manually segmented and generated images?

While manual segmentation is quite tedious and therefore not practical to implement for a large number of datasets, the human eye is very reliable in spotting features. Therefore, manual segmentation is a reasonable ground truth.

7. It appears that all of your quantitative metrics for image segmentation quality are on the training domain and none on held-out samples. It is typical in the ML community to not only compare model accuracy in the training domain, but also in held-out samples that have also been manually segmented. It is common for models to preform very well on the training domain, but less-so on new, held-out data.

No, we do check modal accuracy based on a held-out sample. To make this clear in the maintext, we write (Line 193):

“This manually segmented volume serves as the “ground truth” and is otherwise not used for training. We evaluate segmentation quality according to nineteen standard metrics based on similarity and distance criteria [40], where the learning-based segmentations of one pristine sample are compared to the corresponding “ground truth” (Table S2).”

8. Line 333: Consider reference the work of Franco and coworkers, specifically doi: 10.1016/j.jpowsour.2019.227285.

This is now cited as Reference [53].

9. Line 353: You say that the CBD does not deform extensively. Do you mean it does not plastically/inelastically deform? I suspect that during a charge-discharge cycle, it does deform more significantly.

Yes, we mean that the structure is not deformed. Since this is not an operando experiment, we cannot visualize how the CBD may be changing during a cycle. However, the segmentation could allow us to simulate what is going on.

We now state this more clearly (Line 337): “these values do not change after cycling, highlighting that cycling does not induce a permanent change in the morphology of the carbon black-binder domain in the vicinity of a graphite particle.”

10. You use the phrase “ground truth” to denote the manually segmented image data used for training. Typically, however, “ground truth” refers to training (or validation) data that is known to be exactly correct. Your manual segmentations, while likely good for training, are not perfectly true. You should use “training data” rather than “ground truth”.

We do not use a manually segmented image for training so certainly it should not be called training data. We use the manually segmented image in order to quantify the accuracy of our segmentation. We use “ground truth” in quotations exactly because there is no way of having an exactly correct segmentation.

11. SI Section 3.1: The porosity of 5-40% of CBD seems rather low compared to doi: 10.1002/aenm.201401612.

This is only used for the “basic structure”, CycleGAN takes care of the remaining difference. We now state this explicitly in the SI.

12. SI Section 6: You refer to the output of the network as a “probability map.” You do not have enough details of the network here to verify, but typically these networks output a softmax layer, which is not to be exactly interpreted as a probability map.

We had used the term probability map to make it more understandable for a reader from the battery community, but you are correct that we should be precise. We now use the term “softmax”, also in the main text (Line 305):

*“For each subvolume, a softmax representation is created, which corresponds to a map, where each voxel contains the probability of it belonging to each material phase (pore space, graphite, silicon, or carbon black-binder domain). As described in the **Supplementary Information Note 6**, the probability maps (softmax representations) from each sampling are then combined to obtain the final segmentations (**Figure 6b**).”*

13. Line 354 the authors reference numerical diffusion simulations, but no details on this calculation or the input parameters are given.

We added a detailed description of the performed numerical diffusion simulation in the method section. It now reads:

“The analysis of the twelve segmented datasets is performed in MATLAB using code from Legland et al.[71]. The diffusivity in the through-plane direction was calculated on volumes scaled by a factor of 0.5 (384x384x384 voxel) using the DiffuDict toolbox of the GeoDict2020 Software (Math2Market GmbH, Kaiserslautern, Germany) applying symmetric boundary conditions (Dirichlet). Both border planes (along the through-plane direction) are set to have a constant concentration which decreases edge effects. Symmetric boundary conditions are also applied in all tangential directions and the Neumann (zero flux) condition is used at the domain boundaries. The concentration drop across the through-plane direction was set to 1 and the (dimensionless) Laplace equation is solved in an iterative process by finite volume-based solver (EJ) and stops if the process becomes stationary (tolerance of 0.001).”

14. The methods section is overall very lacking. There are very few details of any of the numerical procedures, which are warranted. Even on the segmentation aspects, more details are required (such as how you did the manual segmentation in Dragonfly).

We thank the reviewer for this suggestion. Since the paper is quite long, we chose to only add a section on the diffusion simulations to the Methods.

We added a description of the manual segmentation with Dragonfly in the SI Section 5: Manual Segmentation. We also refer to the SI in the method section of the main text.

In the SI, it now reads

:

“Three-dimensional manual segmentation is performed with the Dragonfly software on one of the pristine datasets. The different features of interest are identified by the human eye and painted according to the specific material phase. “

Explanations about numerical procedures of generating a digital twin are described in the SI, specifically Section 4: Tomography Simulations, Section 6: Deployment and Section 8: Segmentation Evaluation.

Reviewer #2 (Remarks to the Author): (Part II)

1. A key aspect of this paper, the application of 3D DL algorithms for segmenting volumetric XTM data, is not particularly novel. The approach is well established in the medical literature and is making its way into many application areas. While DL it is not widespread in the field of batteries as of yet, there are recent papers/preprints showing its use on battery electrodes and the extension to the field of batteries is relatively obvious. Commercial software such as Avizo and Dragonfly advertise their software on datasets very similar to this. This is not a criticism of the work itself, which is excellent, but instead a comment on the statements of novelty within the manuscript. What is particularly novel and valuable is the hybrid approach to training the DL segmentation algorithm. The idea of training DL algorithms based on synthetic data has been evolving recently and this manuscript picks up that idea and runs with it, showing its utility.

We thank the reviewer for highlighting the valuability of our work to the community, and we have tried to clearly point this out in the main text (see comments below).

2. I don't fully understand Figure 4. What is the value in the lower row of images (the small "insets" at the bottom of b, d, f, h)? They are insets from the larger images, yet they are not any larger and therefore don't help us to see anything particularly better.

The insets in Figure 4 b,d,f,h show the changes of the different segmentations, highlighting the differences in the graphite and CBD phase specifically. We now state this in the Figure caption.

“The insets in b,d,f and h highlight the improvement of the segmentation especially for the graphite and CBD domain. The overlay (in b) shows the final segmentation result (from h) over the original tomographic image.”

3. You use the phrase “ground truth” to denote the manually segmented image data used for validation. Typically, however, “ground truth” refers to training (or validation) data that is known to be exactly correct, e.g. synthetic data. It is more common, however, to refer to this data as “held out” data or “validation” data. Calling it ground truth presumes it is the right answer. While you have likely put a lot of effort into the manual segmentation, there are likely still inaccuracies or uncertainties in its correctness. Again, I am not criticizing the approach, only the nomenclature.

We agree with the reviewer that ground truth refers to an exactly correct dataset. We use “ground truth” in quotations exactly because there is no way of having an exactly correct segmentation. However, it is the segmentation closest to the ground truth which, in our opinion, validates the use of this term.

4. Please provide images of your manual segmentation (or ground truth) images in Figure 4 so the reader can visually verify that the ground truth segmentation is worthy of validating against.

We performed the manual segmentation on only one pristine sample (see SI Section 5: Manual Segmentation) and the reader can visually verify the validity of our “ground truth” in Figure S7. The overlay in Figure 4b of the original tomographic image and its final segmentation visualizes the quality of the final segmentation result. Therefore, we do not show the manual segmentation result explicitly again at this point.

5. I object to the use of the phrase “digital twin” in the context of the artificial training data. A digital twin is a digital representation of a physical part that can be subjected to the same sorts of environments that the physical part is subjected to but using simulation tools. These models then predict the performance and aging of the part. Instead, what you have here is artificial training data (or a synthetic digital representation). Again, not criticizing the approach, but the nomenclature. Seems like you are trying to use a “hot” phrase to bring more attention to your paper, even though it does not really employ digital twins.

We thank the reviewer for pointing out this misconception and we change the term to either “synthetic structure” or “artificially generated electrodes”. Therefore, we also changed the title of our manuscript to:

“Deep Learning-based Segmentation of Lithium-Ion Battery Microstructures enhanced by artificially generated Electrodes”

6. It took me a couple of readings to realize that the CycleGAN was used to generate realistic looking microstructures, but that they did not have to be in the same particle configuration as the real tomography data. Instead, you use the CycleGAN segmentations as labels and then simulate their tomography data. This is a great approach. However, I feel that it isn’t described clearly and concisely enough in the main manuscript, given that it took me so long to get the point. Additionally, this entire approach seems to be key to the novelty and findings of this manuscript, and more of its details should be found in the methods section of the main manuscript.

We agree with the reviewer that the use of the CycleGAN was not described clearly enough in the main text. We changed the relevant paragraph (Line 108):

“We then use some of the high quality, segmented and labeled data from multimodal imaging as templates for an automatic style transfer algorithm, CycleGAN^{44–46}, which we use to convert the basic structures into realistically looking microstructures. We then create the corresponding tomographic reconstructions of these artificially generated electrodes, incorporating the effects of the beamline and measurement (e.g. energy, resolution, artefacts, noise). These tomographic simulations and the corresponding synthetic datasets form the input-output data pairs that can be used to train the network together with the real datasets.”

7. The column labels in Figure 4 are confusing. The first column is the tomography, fine. Is the second column the output of the neural network trained on real learning data, or is this the training data itself? Similarly for columns three and four – is this the training data or the output from the segmentation model? Where is the training data? And where is the validation data for the Dice score?

Figure 4a shows the cross section as imaged by tomography (raw image data), while Figure 4c shows the segmentation of this exact cross section when the neural network was only trained on real learning data. We see that it fails to reliably distinguish pore space from graphite particles and to segment the fine features of the carbon black-binder domain. Figure 4d shows the segmentation after the neural network was trained with hybrid learning data.

We changed Figure 4 to make this clearer. The respective improvement in the dice scores can be found in the text and in Table 1. The overlay at the bottom of Figure 4b visualizes the agreement of the final segmentation with the original tomographic image. As we manually segmented only one dataset (one of the pristine samples), the dice score (and the other metrics) was evaluated for this specific sample only. The manual segmented dataset can be found in Figure S7.

The training data are the two real learning data sets (shown in Figure 3) or the two real learning datasets additional to the 21 artificial learning datasets (hybrid learning data). The generation of these artificial learning datasets can be seen in Figure 5. For training, 500 volume pairs are sampled from each (real and artificial) dataset.

8. While the novelty of the manuscript is clearly in the segmentation approach, the 12 experimental datasets themselves are incredibly unique and will be valuable to the modeling community in their own right. I look forward to their open publication alongside the manuscript.

Upon publication, the experimental datasets as well as their segmentation will be made available.

9. I do not believe that the softmax layer truly represents a “probability map” as is described beginning around line 307. It is, instead, simply a 3D extension to the logistic function used as an activation layer to the neural network. In some neural networks, the softmax layer does not even necessarily train to 0 and 1, but instead some intermediate value (say 0.1 to 0.8) that are not real probabilities. It would be preferable to simply call it the softmax layer or the activation layer. However, I do not have any issue with the way you are averaging this softmax layer in the overlapping regions, it’s really about semantics.

We had used the term probability map to make it more understandable for a reader from the battery community, but you are correct that we should be precise. We use the term “softmax”, also in the main text (Line 305) and don’t use the term “probability map” anymore.

10. There are two figures labeled “Figure 4” and “Figure 5”. I think I figured out which figure the authors were referring to, but this was confusing.

We are sorry for the confusion.

Figure 4 shows the benefits of artificial learning data for segmentation and compares the segmentations resulting from using different training data (only real learning data, hybrid learning data and hybrid learning data + thresholding).

Figure 5 shows the generation of the artificial learning data, starting from the basic structure (a,b) to the realistic artificial dataset after the style transfer (c,d) and the resulting tomography simulation from this realistic artificial dataset (e,f).

11. The methods section is overall very lacking. There are very few details of any of the numerical procedures, which are warranted. Even on the segmentation aspects, more details are required (such as how you did the manual segmentation in Dragonfly).

See comment 14 in part I.

REVIEWERS' COMMENTS

Reviewer #2 (Remarks to the Author):

I appreciate the authors' detailed responses to both sets of my comments and appropriate modifications of the manuscript. They have sufficiently addressed my concerns and I have no further comments. I recommend its publication as is.